# Evaluated CMI Bounds for Meta Learning: Tightness and Expressiveness

**Fredrik Hellström**
Chalmers University of Technology
Gothenburg, Sweden
`frehells@chalmers.se`

**Giuseppe Durisi**
Chalmers University of Technology
Gothenburg, Sweden
`durisi@chalmers.se`

## Abstract

Recent work has established that the conditional mutual information (CMI) framework of Steinke and Zakynthinou (2020) is expressive enough to capture generalization guarantees in terms of algorithmic stability, VC dimension, and related complexity measures for conventional learning (Harutyunyan et al., 2021, Haghifam et al., 2021). Hence, it provides a unified method for establishing generalization bounds. In meta learning, there has so far been a divide between information-theoretic results and results from classical learning theory. In this work, we take a first step toward bridging this divide. Specifically, we present novel generalization bounds for meta learning in terms of the evaluated CMI (e-CMI). To demonstrate the expressiveness of the e-CMI framework, we apply our bounds to a representation learning setting, with $n$ samples from $\hat{n}$ tasks parameterized by functions of the form $f_i \circ h$. Here, each $f_i \in \mathcal{F}$ is a task-specific function, and $h \in \mathcal{H}$ is the shared representation. For this setup, we show that the e-CMI framework yields a bound that scales as $\sqrt{\mathcal{C}(\mathcal{H})/(n\hat{n}) + \mathcal{C}(\mathcal{F})/n}$, where $\mathcal{C}(\cdot)$ denotes a complexity measure of the hypothesis class. This scaling behavior coincides with the one reported in Tripuraneni et al. (2020) using Gaussian complexity.

## 1 Introduction

Meta learning, sometimes referred to as learning to learn, is a process by which performance on a new machine learning task is increased by using knowledge acquired from separate, but related, tasks [1, 2]. Concretely, the meta learner $\hat{\mathcal{A}}$ has access to training data from several different tasks, which are embedded in a common task environment, and aims to extract information from this data. The goal is to use this information to improve the performance of a base learner $\mathcal{A}$ on a new task from the same task environment. For instance, the task environment can consist of different image classification tasks, and the goal of the meta learner is to learn a shared representation for the tasks or to find suitable hyperparameters for a base learner performing image classification.

As in conventional learning, a central goal in meta learning is to bound the gap between the loss on the training data and the population risk on unseen data. Two current approaches for achieving this goal are: i) to use techniques from classical learning theory to obtain minimax performance guarantees, or ii) to use information-theoretic methods to obtain algorithm-, data- and distribution-dependent guarantees. So far, these two lines of work have evolved largely separately. In this paper, we take some steps toward unifying them. Specifically, we: i) derive new, tighter information-theoretic generalization bounds for meta learning, and ii) demonstrate that these bounds are expressive enough to recover bounds for meta learning from classical learning theory. To concretize the discussion in this introduction, we assume that the meta learner outputs a member $h$ of a function class $\mathcal{H}$ on the

36th Conference on Neural Information Processing Systems (NeurIPS 2022).

basis of $n$ samples from $\hat{n}$ different tasks, and that a base learner selects a member $f$ of a function class $\mathcal{F}$, on the basis of the output of the meta learner and $n$ samples from a given task.

**Classical learning theory for meta learning.** The theoretical analysis of the benefits of meta learning in terms of loss bounds dates back to [3], where the notion of task environment was formally introduced. More recently, for the setting of representation learning, [4, Thm. 5] derived a risk bound that scales as[1] $\sqrt{\mathcal{C}(\mathcal{H})/\hat{n}} + \sqrt{\mathcal{C}(\mathcal{F})/n}$, where $\mathcal{C}(\cdot)$ denotes a complexity measure of the function class. This demonstrates the benefit of meta learning for tasks that share a common environment. Indeed, in the conventional single-task learning scenario, the $n$ samples from a given task need to be used for learning $h$ and $f$ simultaneously, leading to a $\sqrt{\mathcal{C}(\mathcal{H} \times \mathcal{F})/n}$ bound. The bound provided in [4, Thm. 5] was later improved by [5] to a scaling of $\sqrt{\mathcal{C}(\mathcal{H})/(n\hat{n}) + \mathcal{C}(\mathcal{F})/n}$. This improved scaling, where $\mathcal{C}(\mathcal{H})$ decays with the product $n\hat{n}$, confirms the intuition that all of the $n\hat{n}$ samples that are observed are informative at the environment level. Meta learning has also been extensively studied in several special cases. For instance, [6, 7, 8] study a setting with linear features and task mappings, while [9, 10] consider an online convex optimization setting. In this paper, we will mainly focus on the representation learning setting.

**Information-theoretic generalization bounds.** For conventional learning, the study of information-theoretic bounds was initiated by [11, 12], where the average generalization gap of a learning algorithm is bounded in terms of the information that the algorithm reveals about the training data. At its heart, this line of work relies on a change of measure technique that relates the training loss to the population loss. While the first information-theoretic bounds were given in terms of the mutual information between the output of the learning algorithm and the full training data, recent works provide bounds in terms of the *disintegrated* mutual information between the *loss* that the algorithm incurs on a *single* sample pair and a selection variable indicating which sample is used for training, given a supersample containing both the training and test data. These developments are due to the samplewise approach of [13], the disintegration introduced in [14], the evaluated conditional mutual information (e-CMI) notion from [15], and combinations and extensions of these from [16, 17, 18, 19, 20]. This line of work is also intimately related to PAC-Bayesian generalization bounds [21, 22], where the generalization gap, averaged over the learning algorithm, is bounded with high probability over the data in terms of a KL divergence. This is explored further in [23, 24].

**Information-theoretic analysis of meta learning.** Recently, information-theoretic generalization bounds have also been applied to meta learning [25, 26, 27]. In parallel, a PAC-Bayesian analysis of meta learning has also been developed [28, 29, 30, 31, 32, 33]. Generalization bounds obtained via information-theoretic methods have also been used as training objectives in order to improve performance [30, 34]. The quantity of interest in meta learning is the meta-population loss, which is the population loss evaluated on a task that was not observed during the meta learning phase. While this quantity is unknown, the meta learner has indirect information about it through the observed meta-training loss, which is the loss that the meta learner incurs on the training samples from each of the observed tasks during the meta learning phase. The standard approach in the information-theoretic and PAC-Bayesian analysis of meta learning consists of two steps. The first step involves bounding the difference between the meta-training loss and a suitably defined auxiliary loss. The second step involves bounding the difference between the meta-population loss and the auxiliary loss. The two natural candidates for this auxiliary loss are the population loss of an observed task and the training loss for an unobserved task. One of these steps (the first or second, depending on the choice of the auxiliary loss) is purely at the task level, while the other is purely at the environment level. This makes it possible to view each of these steps as a conventional learning problem, so that a standard information-theoretic generalization bound can be applied for each step. By the use of the triangle inequality, the two bounds are then combined to obtain a bound on the meta-population loss in terms of the meta-training loss. We will refer to this procedure as a *two-step* derivation. An alternative approach was recently used by [26], where a *one-step* procedure was employed. Rather than relying on an auxiliary loss, [26] immediately bounds the difference between the meta-population loss and the meta-training loss in terms of a mutual information that captures both task level and environment level

---

[1]In the interest of brevity, we suppress logarithmic factors throughout this section.

dependencies. The environment and task level dependencies can then be obtained by decomposing this mutual information. The resulting bound turns out to have a better scaling with $\hat{n}$ than the two-step bounds. However, the information-theoretic analyses of meta learning reviewed so far do not provide any rigorous characterization of the scaling behavior of the bounds. In particular, the dependence of the information measures on the sample size is typically ignored. This precludes a direct comparison between these information-theoretic bounds and classical learning theory results.

**Contributions.**    Focusing on the meta learning setup, we present novel information-theoretic bounds based on the e-CMI framework and demonstrate how to recover minimax results from classical learning theory via these bounds. Our specific contributions are as follows. In Section 3.1, we derive bounds for the average generalization error in terms of the disintegrated, samplewise e-CMI of the meta learner and base learner: in Theorem 1 and 2, we provide square-root bounds, which are shown to be tighter than results in the literature; in Theorem 3, we derive novel bounds in terms of the binary KL divergence. For low values of the training loss, the binary KL bounds display a more favorable dependence on the number of data samples than the square-root bounds. Next, in Section 3.2, we extend these average bounds to obtain high-probability generalization guarantees. This is necessary to perform comparisons with high-probability bounds from classical learning theory. Finally, in Section 4, we demonstrate the expressiveness of our bounds by applying them to a representation learning setting. Under certain assumptions about the hypothesis classes, we provide upper bounds on the information measures that appear in our bound in terms of complexity measures. The results that we obtain via this procedure display a scaling behavior that coincides with the one reported in [5]. This demonstrates that the e-CMI framework is expressive enough to recover the scaling behavior of generalization guarantees for meta learning obtained via classical learning theory.

## 2    Problem Setup and Notation

We now introduce the meta learning setup that we consider throughout the paper, as well as the necessary notation for stating our results. Similar to [3], we consider a task environment formulation that includes the representation learning setting of [4, 5] as a special case.

Our meta learning setup involves the following quantities. We consider a task distribution $\mathcal{D}$ on the task space $\mathcal{T}$. For a given task $\tau \in \mathcal{T}$, there is a corresponding in-task distribution $\mathcal{D}_\tau$ on the sample space $\mathcal{Z}$. The goal of the meta learner is to output a meta hypothesis $U \in \mathcal{U}$. This is done on the basis of $n$ samples from $\hat{n}$ tasks. Formally, the meta learner is a mapping $\hat{\mathcal{A}} : \mathcal{Z}^{n \times \hat{n}} \times \widehat{\mathcal{R}} \to \mathcal{U}$, where the random variable $\hat{R} \in \widehat{\mathcal{R}}$ captures the potential stochasticity of the learner. The goal of the base learner is to output a hypothesis $W \in \mathcal{W}$, given the output of the meta learner and $n$ samples from a specific task. Formally, the base learner is a mapping $\mathcal{A} : \mathcal{Z}^n \times \mathcal{R} \times \mathcal{U} \to \mathcal{W}$. The random vector $R = (R_1, \ldots, R_{\hat{n}}) \in \mathcal{R}^{\hat{n}}$ has entries that capture the potential stochasticity of each base learner. The entries are independent from the data and assumed to be identically distributed.[2] Here, the spaces $\mathcal{U}$ and $\mathcal{W}$ may be function spaces or parameter spaces, depending on the learning algorithms.

Within each task, the training set for the base learner is randomly formed from a supersample according to the conditional mutual information (CMI) framework of [15]. Specifically, for a given task $\tau$, let $Z^\tau \in \mathcal{Z}^{n \times 2}$ denote the supersample, which is an $n \times 2$ matrix with elements generated independently from $\mathcal{D}_\tau$. For convenience, we index the two columns of $Z^\tau$ by 0 and 1 and the rows by $1, \ldots, n$. The training set $Z_S^\tau$ is formed on the basis of a membership vector $S = (S_1, \ldots, S_n)$, with entries generated independently from a $\text{Bern}(1/2)$ distribution. More precisely, the $j$th element of $Z_S^\tau$ is given by $[Z_S^\tau]_j = Z_{j,S_j}^\tau$, i.e., the $S_j$th element from the $j$th row of $Z^\tau$. Furthermore, we let $-S = (1 - S_1, \ldots, 1 - S_n)$ denote the modulo-2 complement of $S$, which we use to form the test set $Z_{-S}^\tau$, whose $j$th element is given by $[Z_{-S}^\tau]_j = Z_{j,-S_j}^\tau$. With this construction, we randomly assign each sample in the supersample to either the training set or test set with equal probability.

We now describe the meta-supersample $Z$, which contains $2n$ samples from $2\hat{n}$ tasks, as in the meta-learning extension of the CMI framework provided in [35]. Throughout, we let $i \in \{1, \ldots, \hat{n}\}$ denote a task index, $j \in \{1, \ldots, n\}$ denote a sample index, and $k, l \in \{0, 1\}$ denote binary indices

---

[2]While identical distributions are not necessary for our results, this assumption simplifies the presentation.

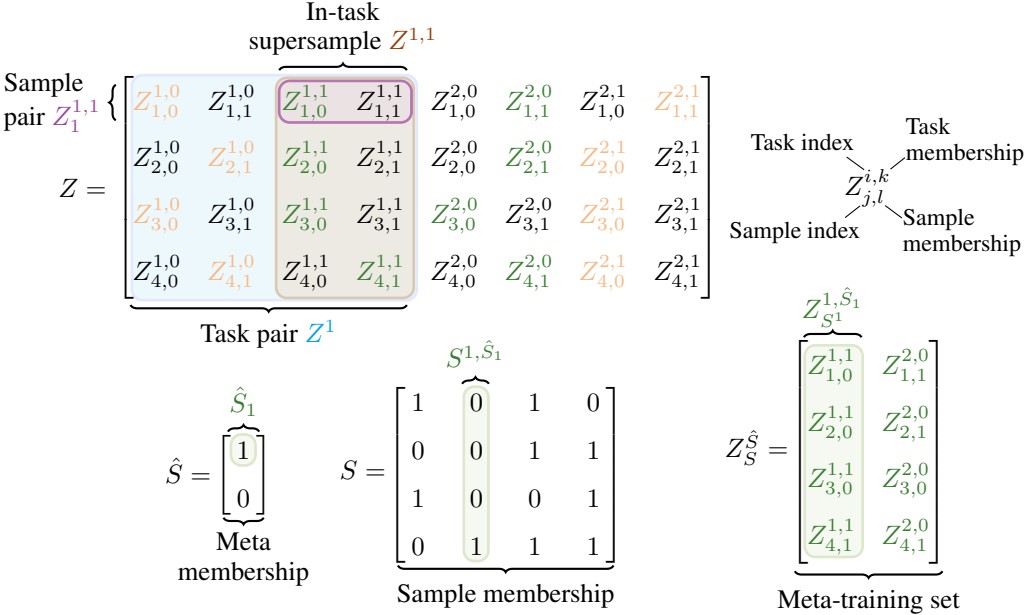

Figure 1: A graphical representation of our notation. In this example, the meta-supersample contains two task pairs: $Z^1$, which is marked in blue, and $Z^2$. In turn, $Z^1$ consists of the two in-task supersamples $Z^{1,0}$ and $Z^{1,1}$, which is marked in brown. Next, $Z^{1,1}$ consists of the four sample pairs $Z_j^{1,1}$, where $j = 1, \ldots, 4$. In the figure, $Z_1^{1,1}$ is marked in purple. Finally, $Z_1^{1,1}$ consists of a pair of samples, $Z_{1,0}^{1,1}$ and $Z_{1,1}^{1,1}$. When a binary vector is used as subscript or superscript, this indicates that we should enumerate the meta-supersample according to that vector. To illustrate this, consider the construction of the meta-training set $Z_S^{\hat{S}}$. The meta-training set is formed on the basis of the meta-subset choice $\hat{S}$ and the observed-task subset choice $S^{\hat{S}}$. For instance, from the first task-pair, $\hat{S}_1 = 1$ indicates that we should select task 1. Then, from the first sample pair in this task, $S^{1,\hat{S}_1} = 0$ indicates that we should select sample 0, which is $Z_{1,0}^{1,1}$, marked in green. Repeating this for each sample pair in the in-task supersample $Z^{1,1}$, we can identify the remaining elements of $Z_{S^1}^{1,\hat{S}_1}$. This procedure is performed for each task index until all samples of $Z_S^{\hat{S}}$ are identified. The meta-test set $Z_{-S}^{-\hat{S}}$ is formed by an analogous procedure, but is now based on $-\hat{S}$ and $-S^{-\hat{S}}$. The entries from the meta-supersample that are selected for $Z_{-S}^{-\hat{S}}$ are marked in orange.

indicating task membership and sample membership respectively. Formally, the meta-supersample $Z$ can be viewed as a data structure with $4n\hat{n}$ elements. In Figure 1, we illustrate $Z$ as an $n \times 4\hat{n}$ matrix for the case of $\hat{n} = 2$ task pairs and $n = 4$ sample pairs for each task. We decompose $Z$ as $(Z^1, \ldots, Z^{\hat{n}})$, where each element can be seen as a task pair. Specifically, the pair $Z^i$ can be decomposed as $(Z^{i,0}, Z^{i,1})$, where each element is a task-specific supersample as described above. The task-specific supersamples are $Z^{i,k} = (Z_1^{i,k}, \ldots, Z_n^{i,k})$, where each element is a pair of data samples. Specifically, each sample pair is $Z_j^{i,k} = (Z_{j,0}^{i,k}, Z_{j,1}^{i,k})$, where $Z_{j,l}^{i,k} \in \mathcal{Z}$. The elements of $Z$ are generated as follows. First, we generate $\tau_{i,k} \sim \mathcal{D}$. Then, we independently generate the samples $Z_{j,l}^{i,k} \sim \mathcal{D}_{\tau_{i,k}}$. This is repeated for all indices to form $Z$.

Finally, we describe how the meta-training data is selected from the meta-supersample $Z$. This is done on the basis of the meta-membership vector $\hat{S}$ and the in-task membership vector $S$. Specifically, the meta-membership vector $\hat{S} = (\hat{S}_1, \ldots, \hat{S}_{\hat{n}})$ is an $\hat{n}$-dimensional vector, while the sample membership vector $S = (S^{1,0}, S^{1,1}, \ldots, S^{\hat{n},0}, S^{\hat{n},1})$ is a collection of $2\hat{n}$ vectors, where each $S^{i,k} = (S_1^{i,k}, \ldots, S_n^{i,k})$ is an $n$-dimensional vector. The elements of all these vectors are generated independently from a $\mathrm{Bern}(1/2)$ distribution. For any Bernoulli matrix $X$, we let $-X$ denote its elementwise complement modulo 2, i.e., $\mathbf{1} - X$, where $\mathbf{1}$ is the all-one matrix.

These membership vectors are used to form the meta-training set as follows. We use the convention that, when a binary vector is used as a subscript or superscript of $Z$, this indicates that we should enumerate over this vector. Using this convention, the training set for the $(i,k)$th task is constructed as $Z_{S^{i,k}}^{i,k} = (Z_{1,S_1^{i,k}}^{i,k}, \ldots, Z_{n,S_n^{i,k}}^{i,k})$. We will use the shorthands $Z_{S^i}^{i,k} = Z_{S^{i,k}}^{i,k}$ and $Z_{1,S_j^i}^{i,k} = Z_{1,S_j^{i,k}}^{i,k}$. The test set for the $(i,k)$th task $Z_{-S^i}^{i,k}$ is constructed analogously, but on the basis of $-S^{i,k}$. The full meta-training set is $Z_S^{\hat{S}} = (Z_{S^{1,\hat{S}_i}}^{1,\hat{S}_1}, \ldots, Z_{S^{\hat{n},\hat{S}_{\hat{n}}}}^{\hat{n},\hat{S}_{\hat{n}}})$, and the meta-test set $Z_{-S}^{-\hat{S}}$ is defined analogously. With this construction, each task in the meta-supersample is assigned to either the meta-training set or the meta-test set with equal probability. Then, as before, the samples within each task are assigned to an in-task training set or test set with equal probability. The meta-training set consists of training samples within training tasks, while the meta-test set consists of test samples within test tasks.

We denote the output of the meta learner as $U = \hat{\mathcal{A}}(Z_S^{\hat{S}}, \hat{R}) \in \mathcal{U}$ and the output of the base learner for task $(i,k)$ as $W^{i,k} = \mathcal{A}(Z_{S^i}^{i,k}, R_i, U) \in \mathcal{W}$. The performance of the learners is evaluated through a loss function $\ell : \mathcal{W} \times \mathcal{Z} \to [0,1]$. We denote the losses that the meta learner and base learner induce on the meta-supersample $Z$ by $\lambda$, which inherits the subscript and superscript notation that we described for $Z$. Thus, we have $\lambda_{j,l}^{i,k} = \ell(W^{i,k}, z_{j,l}^{i,k})$. In other words, $\lambda_{j,l}^{i,k}$ is the loss induced on the $(j,l)$th sample in the $(i,k)$th task.

On the basis of the loss matrix $\lambda$ and the membership vectors $\hat{S}$ and $S$, we can compute four different losses. The main quantity that we are interested in bounding is the average meta-population loss $L_{\mathcal{D}}$, which is the loss on test data for unobserved tasks. The quantity that the meta learner has access to is the average meta-training loss, $\widehat{L}$, which is the training loss for observed tasks. The other two losses are the average auxiliary test loss, $\bar{L}$, which is the loss on test data for observed tasks, and the average auxiliary training loss $\widetilde{L}$, which is the loss on training data for unobserved tasks. In the two-step derivations, one of these two quantities is used as the auxiliary loss. These four losses are given by

$$L_{\mathcal{D}} = \frac{1}{n\hat{n}} \sum_{i,j=1}^{\hat{n},n} \mathbb{E}_{\lambda_j^i, Z}\left[ \lambda_{j,-S_j^i}^{i,-\hat{S}_i} \right], \qquad \widehat{L} = \frac{1}{n\hat{n}} \sum_{i,j=1}^{\hat{n},n} \mathbb{E}_{\lambda_j^i, Z}\left[ \lambda_{j,S_j^i}^{i,\hat{S}_i} \right] \qquad (1)$$

$$\bar{L} = \frac{1}{n\hat{n}} \sum_{i,j=1}^{\hat{n},n} \mathbb{E}_{\lambda_j^i, Z}\left[ \lambda_{j,-S_j^i}^{i,\hat{S}_i} \right], \qquad \widetilde{L} = \frac{1}{n\hat{n}} \sum_{i,j=1}^{\hat{n},n} \mathbb{E}_{\lambda_j^i, Z}\left[ \lambda_{j,S_j^i}^{i,-\hat{S}_i} \right]. \qquad (2)$$

Finally, we end this section by introducing some information-theoretic quantities that appear in our bounds. First, let $P$ and $Q$ be two probability measures such that $P$ is absolutely continuous with respect to $Q$. The KL divergence between $P$ and $Q$ is denoted by $D(P \,\|\, Q)$. For the special case where $P$ and $Q$ are Bernoulli distributions with parameters $p$ and $q$, we let

$$d(p \,\|\, q) = D(P \,\|\, Q) = p \log\left(\frac{q}{p}\right) + (1-p) \log\left(\frac{1-p}{1-q}\right). \qquad (3)$$

We refer to $d(p \,\|\, q)$ as the binary KL divergence. The mutual information between the random variables $X$ and $Y$ is given by $I(X;Y) = D(P_{XY} \,\|\, P_X P_Y)$, where $P_{XY}$ is the joint distribution of $X$ and $Y$ and $P_X$ and $P_Y$ are the corresponding marginals. The disintegrated mutual information between $X$ and $Y$ given a third random variable $Z$ is given by $I^Z(X;Y) = D(P_{XY|Z} \,\|\, P_{X|Z}P_{Y|Z})$, where $P_{XY|Z}$ is the conditional joint distribution of $X$ and $Y$ given $Z$ and $P_{X|Z}P_{Y|Z}$ is the product distribution formed from the corresponding marginals. The expectation over $Z$ of the disintegrated mutual information is the conditional mutual information $I(X;Y|Z) = \mathbb{E}_Z\big[ I^Z(X;Y) \big]$.

## 3 Generalization Bounds for Meta Learning with e-CMI

In this section, we present generalization bounds in terms of the e-CMI of the meta learner and base learner. In Section 3.1, we derive average square-root bounds that tighten results from [26, 35], as well as novel binary KL bounds. In Section 3.2, we extend these results to obtain novel, high-probability information-theoretic bounds for meta learning. In Section 4, we demonstrate the expressiveness of

the e-CMI framework by using the bounds from this section to recover generalization guarantees from classical learning theory for representation learning.

## 3.1 Average Bounds

In Theorem 1, we present a square-root bound for the average generalization error obtained through a two-step derivation. Specifically, one step consists of bounding the unobserved training loss $\widetilde{L}$ in terms of the observed training loss $\widehat{L}$, and the second step bounds the meta-population loss $L_{\mathcal{D}}$ in terms of $\widetilde{L}$. Chaining these two bounds, we obtain a bound on $L_{\mathcal{D}}$ in terms of $\widehat{L}$. The bound depends on the information captured by two random variables: the task-level variable $\lambda_j^{i,-\hat{S}_i}$, which contains the training loss and test loss for task $(i, -\hat{S}_i)$, as well as the environment-level variable $\lambda_{j,S_j^i}^i$, which contains the training losses for both the observed task $(i, \hat{S}_i)$ and the unobserved task $(i, -\hat{S}_i)$. We provide the proof of this result in Appendix A, along with the proofs of all other results in this paper.

**Theorem 1** (Two-step square-root bound). *Consider the setup described in Section 2. Then,*

$$\left| L_{\mathcal{D}} - \widehat{L} \right| \leq \frac{1}{n\hat{n}} \sum_{i,j=1}^{\hat{n},n} \mathbb{E}_{Z,S_j^i} \left[ \sqrt{2 I^{Z,S_j^i}(\lambda_{j,S_j^i}^i; \hat{S}_i)} \right] + \frac{1}{n\hat{n}} \sum_{i,j=1}^{\hat{n},n} \mathbb{E}_{Z,\hat{S}_i} \left[ \sqrt{2 I^{Z,\hat{S}_i}(\lambda_j^{i,-\hat{S}_i}; S_j^{i,-\hat{S}_i})} \right]. \quad (4)$$

The first term captures the environment-level generalization error while the second term captures the task-level generalization error. In order to clarify the relation between Theorem 1 and results from the literature, we relax it by upper-bounding the disintegrated individual-sample e-CMI terms by their integrated, full-sample, parametric CMI counterparts.

**Corollary 1.** *Theorem 1 implies that*

$$\left| L_{\mathcal{D}} - \widehat{L} \right| \leq \sqrt{\frac{2 I(U; \hat{S} | Z, S)}{\hat{n}}} + \sqrt{\frac{2 I(W^{1,-\hat{S}_1}; S^{1,-\hat{S}_1} | Z, \hat{S}_1)}{n}}. \quad (5)$$

This recovers the result of [35, Thm. 1], demonstrating that Theorem 4 is tighter.

Next, we present an alternative square-root bound that is obtained through a one-step derivation. This bound depends on the information captured by the random variable $\lambda_j^i$, which contains the training and test loss for both the observed and unobserved tasks.

**Theorem 2** (One-step square-root bound). *Consider the setup described in Section 2. Then,*

$$\left| L_{\mathcal{D}} - \widehat{L} \right| \leq \frac{1}{n\hat{n}} \sum_{i,j=1}^{\hat{n},n} \mathbb{E}_Z \left[ \sqrt{2 I^Z(\lambda_j^i; \hat{S}_i, S_j^i)} \right]. \quad (6)$$

Again, to compare this bound to results in the literature, we relax it by upper-bounding the disintegrated individual-sample e-CMI terms by their integrated, full-sample, parametric counterparts.

**Corollary 2.** *Let $W^i = (W^{i,0}, W^{i,1})$ and $W = \{W^i\}_{i=1}^{\hat{n}}$. Then,*

$$\left| L_{\mathcal{D}} - \widehat{L} \right| \leq \sqrt{\frac{2 I(W; \hat{S}, S | Z)}{n\hat{n}}} \leq \sqrt{\frac{2 I(U; \hat{S}, S | Z) + 2\hat{n} I(W^1; S^1 | Z, U)}{n\hat{n}}} \quad (7)$$

$$\leq \sqrt{\frac{2 I(U; Z_S^{\hat{S}}) + 2\hat{n} I(W^1; Z_{S^1}^1 | U)}{n\hat{n}}}. \quad (8)$$

Up to some constant factors, this recovers the result in [26, Thm. 5.1]. Note that, if $\hat{\mathcal{A}}$ or $\mathcal{A}$ are deterministic learning algorithms with continuous outputs, the mutual information terms in (8) are unbounded. In contrast, the CMI terms in (7) are always finite. This is discussed in more detail in [15]. Furthermore, the bound in (7) compares favorably to [35, Thm. 1], since it decays with the product $n\hat{n}$ rather than with $n$ and $\hat{n}$ separately. This improvement is due to the one-step derivation.

Finally, in Theorem 3, we present two novel bounds in terms of the binary KL divergence. The advantage of these bounds, as compared to the square-root bounds in Theorem 1 and 2, is that they have a more favorable dependence on the number of samples for low training losses. We demonstrate this improved rate for representation learning in Section 4.

**Theorem 3** (Binary KL bounds). *For $m \geq 2$, $q \in [0, 1]$ and $c > 0$, let*

$$d_m^{-1}(q, c) = \sup \left\{ p \in [0, 1] : d\left( q \,||\, \frac{q+p}{m} \right) \leq c \right\}. \tag{9}$$

*Then,*

$$L_{\mathcal{D}} \leq d_2^{-1} \left( d_2^{-1} \left( \widehat{L}, \frac{1}{n\hat{n}} \sum_{i,j=1}^{\hat{n},n} I(\lambda_{j,S_j^i}^i; \hat{S}_i | Z, S_j^i) \right), \frac{1}{n\hat{n}} \sum_{i,j=1}^{\hat{n},n} I(\lambda_j^{i,-\hat{S}_i}; S_j^{i,-\hat{S}_i} | Z, \hat{S}_i) \right). \tag{10}$$

*Furthermore,*

$$L_{\mathcal{D}} + \bar{L} + \widetilde{L} \leq d_4^{-1} \left( \widehat{L}, \frac{1}{n\hat{n}} \sum_{i,j=1}^{\hat{n},n} I(\lambda_j^i; \hat{S}_i, S_j^i | Z) \right). \tag{11}$$

Interestingly, (11) provides a bound on the sum of the average meta-population loss $L_{\mathcal{D}}$, the test loss on observed tasks $\bar{L}$, and the training loss on unobserved tasks $\widetilde{L}$. Due to the nonnegativity of the loss, we can obtain an explicit bound on $L_{\mathcal{D}}$ by using the lower bound $\bar{L} + \widetilde{L} \geq 0$, which is a sensible relaxation when $L_{\mathcal{D}}$ is the dominant term. By this relaxation, we weaken the bound at most by a constant factor. As previously mentioned, the bounds in Theorem 3 can have a more favorable dependence on the number of samples than the square-root bounds in Theorem 1 and 2 when the training loss is low. In the following corollary, we present a bound on $L_{\mathcal{D}}$ for the case where $\widehat{L} = 0$.

**Corollary 3.** *Assume that $\widehat{L} = 0$. Then, Theorem 3 implies that*

$$L_{\mathcal{D}} \leq 4 - 4 \exp \left( -\frac{1}{n\hat{n}} \sum_{i,j=1}^{\hat{n},n} I(\lambda_j^i; \hat{S}_i, S_j^i | Z) \right) \leq \frac{4}{n\hat{n}} \sum_{i,j=1}^{\hat{n},n} I(\lambda_j^i; \hat{S}_i, S_j^i | Z). \tag{12}$$

Compared to the bound in Theorem 2, there is no square root in Corollary 3. As we show in Section 4, this can lead to a faster rate of decay with the number of samples.

## 3.2 High-probability Bounds

In the previous section, we provided bounds on the average generalization error. However, meta learning bounds obtained via classical learning theory are typically high-probability bounds [4, 5]. In order to assess the expressiveness of the e-CMI framework in terms of its ability to recover these results, we now extend the bounds from Section 3.1 to the high-probability setting. For this, we need some additional notation. We let $L_{\mathcal{D}}(Z, \hat{S}, S)$ and $\widehat{L}(Z, \hat{S}, S)$ denote the meta-population loss and training loss given that the meta-training set is constructed from $(Z, \hat{S}, S)$. Specifically,

$$L_{\mathcal{D}}(Z, \hat{S}, S) = \frac{1}{n\hat{n}} \sum_{i,j=1}^{\hat{n},n} \mathbb{E}_{\hat{R}, R_i} \left[ \ell(\mathcal{A}(Z_{S^i}^{i,-\hat{S}_i}, R_i, \hat{\mathcal{A}}(Z_S^{\hat{S}}, \hat{R})), Z_{j,-S_j^i}^{i,-\hat{S}_i}) \right], \tag{13}$$

$$\widehat{L}(Z, \hat{S}, S) = \frac{1}{n\hat{n}} \sum_{i,j=1}^{\hat{n},n} \mathbb{E}_{\hat{R}, R_i} \left[ \ell(\mathcal{A}(Z_{S^i}^{i,\hat{S}_i}, R_i, \hat{\mathcal{A}}(Z_S^{\hat{S}}, \hat{R})), Z_{j,S_j^i}^{i,\hat{S}_i}) \right]. \tag{14}$$

We now present a high-probability version of the two-step square root bound in Theorem 1. To simplify the presentation, we omit explicit constants and assume that $\hat{\mathcal{A}}$ and $\mathcal{A}$ are indifferent to the order of the data samples. The theorem statement is provided in more general form in Appendix A.

**Theorem 4** (High-probability two-step square-root bound). *Let $Q_{1,S_1}$ denote the conditional distribution of $\lambda_{1,S_1}$ given $(Z, \hat{S}, S)$, and let $P_{1,S_1}$ denote $\mathbb{E}_{\hat{S}}[Q_{1,S_1}]$. Furthermore, let $Q^{1,-\hat{S}_1}$ denote the conditional distribution of $\lambda^{1,-\hat{S}_1}$ given $(Z, \hat{S}_1, S^{1,-\hat{S}_1})$, and let $P^{1,-\hat{S}_1}$ denote $\mathbb{E}_{S^{1,-\hat{S}_1}}\left[Q^{1,-\hat{S}_1}\right]$. Then, there exist constants $C_1, C_2$ such that, with probability at least $1-\delta$ under the draw of $(Z, \hat{S}, S)$,*

$$\left|L_{\mathcal{D}}(Z, \hat{S}, S) - \widehat{L}(Z, \hat{S}, S)\right| \le C_1 \sqrt{\frac{D(Q_{1,S_1} \,\|\, P_{1,S_1}) + \log(\frac{n\sqrt{\hat{n}}}{\delta})}{\hat{n}}}$$
$$+ C_2 \sqrt{\frac{D(Q^{1,-\hat{S}_1} \,\|\, P^{1,-\hat{S}_1}) + \log(\frac{\sqrt{n}}{\delta})}{n}}. \quad (15)$$

The KL divergences in (15) can be interpreted as pointwise e-CMIs. Indeed,

$$\mathbb{E}_{Z,\hat{S},S}[D(Q_{1,S_1} \,\|\, P_{1,S_1})] = I(\lambda_{1,S_1}; \hat{S}|S, Z), \quad (16)$$

$$\mathbb{E}_{Z,\hat{S},S}\left[D(Q^{1,-\hat{S}_1} \,\|\, P^{1,-\hat{S}_1}))\right] = I(\lambda^{1,-\hat{S}_1}; S^{1,-\hat{S}_1}|Z, \hat{S}_1). \quad (17)$$

Finally, we present a high-probability version of the one-step square root bound in Theorem 2.

**Theorem 5** (High-probability one-step square-root bound). *Let $Q$ denote the conditional distribution of $\lambda$ given $(Z, \hat{S}, S)$, and let $P$ denote $\mathbb{E}_{\hat{S},S}[Q]$. Then, with probability at least $1 - \delta$ under the draw of $(Z, \hat{S}, S)$,*

$$\left|L_{\mathcal{D}}(Z, \hat{S}, S) - \widehat{L}(Z, \hat{S}, S)\right| \le \sqrt{\frac{2\left(D(Q \,\|\, P) + \log(\frac{\sqrt{n\hat{n}}}{\delta})\right)}{n\hat{n} - 1}}. \quad (18)$$

Again, the KL divergence can be interpreted as a pointwise e-CMI, since

$$\mathbb{E}_{Z,\hat{S},S}[D(Q \,\|\, P)] = I(\lambda; \hat{S}, S|Z). \quad (19)$$

## 4 Expressiveness of the Bounds

In Section 3, we presented several new information-theoretic generalization bounds, and demonstrated that they improve upon known bounds from the literature. We now turn our focus to the expressiveness of the e-CMI framework. In particular, we show that the bounds from Section 3 can be used to recover generalization guarantees for meta learning from classical learning theory. Specifically, we consider the representation learning setting that is analyzed in [5]. We use the following notation. First, the sample space is the product of an instance space and label space: $\mathcal{Z} = \mathcal{X} \times \mathcal{Y}$. The aim of the meta learner is to find a representation $h_U \in \mathcal{H}$, while the base learner outputs a task-specific function $f_W \in \mathcal{F}$. Composing these functions, we obtain the mapping $f_W \circ h_U : \mathcal{X} \to \mathcal{Y}$.

### 4.1 Minimax Generalization Bounds

To obtain explicit minimax bounds, we assume that $\mathcal{H}$ has finite Natarajan dimension $d_N$ and that $\mathcal{F}$ has finite VC dimension $d_{\text{VC}}$. This allows us to derive bounds on the entropy of the representations and predictions that the meta learner and base learner induce on the meta-supersample. This, in turn, leads to bounds on the e-CMI terms that appear in the bounds in Section 3. In the following corollary, we present the bounds that are obtained by bounding the e-CMI terms in Theorem 1 and 2. These hold for any learner that outputs hypotheses from the specified classes.

**Corollary 4.** *Assume that the range of $\mathcal{H}$ has cardinality $N$, that the Natarajan dimension of $\mathcal{H}$ is $d_N$, and that the VC dimension of $\mathcal{F}$ is $d_{\text{VC}}$. Also, let $2n \ge d_{\text{VC}} + 1$ and $2\hat{n} \ge d_N + 1$. Then,*

$$\left|L_{\mathcal{D}} - \widehat{L}\right| \le \sqrt{\frac{2d_N \log\left(\binom{N}{2}\frac{2e\hat{n}}{d_N}\right)}{\hat{n}}} + \sqrt{\frac{2d_{\text{VC}} \log\left(\frac{2en}{d_{\text{VC}}}\right)}{n}}, \quad (20)$$

$$\left| L_\mathcal{D} - \widehat{L} \right| \leq \sqrt{\frac{2d_N \log\left(\binom{N}{2}\frac{4en\hat{n}}{d_N}\right) + 4\hat{n}d_{\mathrm{VC}} \log\left(\frac{2en}{d_{\mathrm{VC}}}\right)}{n\hat{n}}}. \tag{21}$$

Corollary 4 establishes that, for the average setting, we can use the bounds in Theorems 1 and 2 to obtain minimax bounds for function classes with bounded Natarajan and VC dimensions. Note that, in the upper-bound of (20), we have fully decoupled the complexity of the two function classes. This is made possible by the fact that we used $\widetilde{L}$ as the auxiliary loss in the derivation of Theorem 1, rather than $\bar{L}$. We discuss this in more detail in Appendix A.

Next, we consider the interpolating setting, where $\widehat{L} = 0$. Under this assumption, we demonstrate that we can achieve a better rate of convergence with respect to the number of training samples. The result, presented in the following corollary, relies on similarly bounding the e-CMI term in Corollary 3.

**Corollary 5.** *Consider the setting of Corollary 4. Furthermore, assume that $\widehat{L} = 0$. Then,*

$$L_\mathcal{D} \leq \frac{4d_N \log\left(\binom{N}{2}\frac{4en\hat{n}}{d_N}\right) + 8\hat{n}d_{\mathrm{VC}} \log\left(\frac{2en}{d_{\mathrm{VC}}}\right)}{n\hat{n}}. \tag{22}$$

The result in Corollary 5 demonstrates that, for the interpolating setting, the e-CMI framework is expressive enough to yield a bound that, ignoring logarithmic factors, decays as $1/(n\hat{n})$, often referred to as a fast rate.

Finally, noting that the bounds in [4] and [5] are high-probability rather than average bounds, we also derive high-probability generalization bounds. In order to achieve this, we need probabilistic upper bounds on the KL divergences that appear in Theorem 4 and 5, similar to how the e-CMI terms were bounded for Corollary 4 and 5. The resulting bounds are presented in the following corollary.

**Corollary 6.** *Consider the setting of Corollary 4. Then, there exist constants $C_1, C_2, C_3$ such that, with probability at least $1 - \delta$ under the draw of $(Z, \hat{S}, S)$,*

$$\left| L_\mathcal{D}(Z,\hat{S},S) - \widehat{L}(Z,\hat{S},S) \right| \leq C_1 \sqrt{\frac{d_N \log\left(\binom{N}{2}\frac{\hat{n}}{d_N}\right) + \log\left(\frac{n\sqrt{\hat{n}}}{\delta}\right)}{\hat{n}}} + C_2 \sqrt{\frac{d_{\mathrm{VC}} \log\left(\frac{n}{d_{\mathrm{VC}}}\right) + \log\left(\frac{\sqrt{n}}{\delta}\right)}{n}}, \tag{23}$$

$$\left| L_\mathcal{D}(Z,\hat{S},S) - \widehat{L}(Z,\hat{S},S) \right| \leq C_3 \sqrt{\frac{d_N \log\left(\binom{N}{2}\frac{n\hat{n}}{d_N}\right) + \hat{n}d_{\mathrm{VC}} \log\left(\frac{n}{d_{\mathrm{VC}}}\right) + \log\left(\frac{\sqrt{n\hat{n}}}{\delta}\right)}{n\hat{n}}}. \tag{24}$$

We now see that, suppressing logarithmic factors, the upper bound in (25) scales as $\sqrt{\mathcal{C}(\mathcal{H})/\hat{n}} + \sqrt{\mathcal{C}(\mathcal{F})/n}$, whereas the upper bound in (24) scales as $\sqrt{\mathcal{C}(\mathcal{H})/(n\hat{n}) + \mathcal{C}(\mathcal{F})/n}$. This matches the rates obtained by [4] and [5], respectively, demonstrating that the e-CMI framework, combined with the one-step approach, is expressive enough to recover the scaling of these results.

Note that there are some differences between these results and the ones in [4, 5]. First, while the complexity measures that we use are related to the Natarajan and VC dimension, the results in [5] are given in terms of Gaussian complexity. Furthermore, while [4, 5] provide excess risk bounds for a fixed target task with $m$ training samples, the bounds in Corollary 6 are generalization bounds for a randomly drawn task. In Section 4.2, we extend our analysis to derive excess risk bounds for a fixed target task.

## 4.2   Excess Risk Bounds

In order to derive excess risk bounds for a specific target task, as is done in [5], we need to assume that the meta learner $\hat{\mathcal{A}}$ and the base learner $\mathcal{A}$ are empirical risk minimizers. This is in contrast to all previous bounds in this paper, which apply to any learning algorithms. Furthermore, we need a notion of oracle algorithms, which minimize the population loss. Finally, we need to assume that the tasks contained in the meta-supersample satisfy a notion of *task diversity*. Intuitively, this means that, given the output of the empirical risk-minimizing meta learner, the performance of the oracle base

learner on the tasks in the meta-supersample gives a reasonable indication of the performance of the oracle base learner on any possible task. Due to space constraints, we state here an informal version of a high-probability excess risk bound for a specified target task based on the one-step square-root generalization bound in Corollary 6. A precise statement of this result, along with its proof, is given in Appendix B.

**Corollary 7** (Informal). *Consider the setting of Corollary 6 and a fixed task $\tau_0$. Let $Z_{S^0}^0 \in \mathcal{Z}^m$ be a vector of $m$ samples generated independently according to the data distribution $\mathcal{D}_{\tau_0}$ for task $\tau_0$. Let $\hat{\mathcal{A}}$ and $\mathcal{A}$ be empirical risk minimizers. Let $L_0(Z, \hat{S}, S, Z_{S^0}^0)$ denote the population loss on task $\tau_0$ when applying $\hat{\mathcal{A}}$ to $Z_S^{\hat{S}}$ and $\mathcal{A}$ to $(Z_{S^0}^0, \hat{\mathcal{A}}(Z_S^{\hat{S}}))$, and let $L_0^*$ denote the smallest population loss for task $\tau_0$ that can be obtained using functions from $\mathcal{F}$ and $\mathcal{H}$. Finally, assume that the supersample satisfies a task-diversity assumption with parameters $\nu, \epsilon$. Then, there exist constants $C_1$ and $C_2$ such that, with probability at least $1 - \delta$ under the draw of $(Z, \hat{S}, S, Z_{S^0}^0)$,*

$$L_0(Z, \hat{S}, S, Z_{S^0}^0) - L_0^* \leq C_1 \sqrt{\frac{d_{\mathrm{VC}} \log\left(\frac{\sqrt{m}}{d_{\mathrm{VC}}}\right) + \log\left(\frac{\sqrt{m}}{\delta}\right)}{m}}$$
$$+ C_2 \nu^{-1} \sqrt{\frac{d_N \log\left(\binom{N}{2} \frac{n\hat{n}}{d_N}\right) + \hat{n} d_{\mathrm{VC}} \log\left(\frac{n}{d_{\mathrm{VC}}}\right) + \log\left(\frac{\sqrt{n\hat{n}}}{\delta}\right)}{n\hat{n}}} + \epsilon. \quad (25)$$

The bound in (25) displays the same scaling as the excess risk bound in [5].

It is possible to derive high-probability bounds based on the two-step square-root generalization bound in Corollary 6 by suitably substituting the two-step bound in the proof of Corollary 7. The same can be done using the bounds that are given in terms of information measures, and average excess risk bounds can also be derived by an analogous procedure. Finally, we note that it is possible to derive excess risk bounds for a new, random task, rather than a specified target task, without assuming task diversity. This is done in Corollary 8 in Appendix B.

## 5  Conclusions

In this paper, we derived new generalization bounds for meta learning using e-CMI, which improve upon information-theoretic bounds found in the literature. By considering a representation learning setting, we demonstrated that e-CMI bounds obtained via a conventional two-step approach lead to rates that coincide with those found in [4]. In contrast, we showed that by combining the e-CMI framework with a one-step approach, we recover the more favourable scaling found in [5]. Note that, while the bounds in [5] are uniform over the hypothesis class, the information-theoretic bounds that we derive are inherently algorithm- and data-dependent. As a consequence, they are nonvacuous when applied to settings such as classification with deep neural networks [20]. The algorithm-dependence and expressiveness of our bounds indicate that they can be developed further to guide algorithm design. However, no recipe for this is provided in this paper. It should also be noted that the complexity measures that we consider differ from the Gaussian complexity in [5]. An intriguing topic for further study is to clarify the connection between e-CMI and Gaussian complexity.

## Acknowledgements

This work was partly supported by the Wallenberg AI, Autonomous Systems and Software Program (WASP) funded by the Knut and Alice Wallenberg Foundation and the Chalmers AI Research Center (CHAIR).

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
