# A Proofs

In this appendix, we present the proofs of the results in the main paper. First, we give a summary of the notation that is used in this appendix. Then, in Section A.1, we present some lemmas that are useful for proving our main results. In Section A.2, we prove the average generalization bounds from Section 3.1. In Section A.3, we prove the high-probability results from Section 3.2. Finally, in Section A.4, we prove the generalization bounds for multiclass classification from Section 4.1.

**Notation summary.** For $i \in \{1, \ldots, \hat{n}\}$, $k \in \{0, 1\}$, $j \in \{1, \ldots, n\}$, and $l \in \{0, 1\}$, we let $Z_{j,l}^{i,k}$ denote the $l$th sample from the $j$th sample pair in the $k$th task of the $i$th task pair. This is illustrated in Figure 1. Throughout, $i$ denotes a task index, $j$ denotes a sample index, $k$ denotes a selection within the task pair, and $l$ denotes a selection within the sample pair. Furthermore, we let $Z_{j,l}^i = (Z_{j,l}^{i,0}, Z_{j,l}^{i,1})$ and $Z_j^i = \{Z_{j,l}^{i,k}\}_{l=0,1}^{k=0,1}$. The tasks used to form the training set are selected on the basis of the binary vector $\hat{S} = (\hat{S}_1, \ldots, \hat{S}_{\hat{n}})$. Within task $(i, k)$, the samples that form the training set are selected on the basis of $S^{i,k} = (S_1^{i,k}, \ldots, S_n^{i,k})$. For convenience, we let $S^i = (S^{i,0}, S^{i,1})$ and $S = (S^1, \ldots, S^{\hat{n}})$. The training set for task $(i, k)$ is $Z_{S^{i,k}}^{i,k} = (Z_{1,S_1^{i,k}}^{i,k}, \ldots, Z_{n,S_n^{i,k}}^{i,k})$. As a shorthand, $Z_{S^i}^{i,k} = Z_{S^{i,k}}^{i,k}$. The collection of all samples is $Z = \{Z_j^i\}_{j=1:n}^{i=1:\hat{n}}$. The full data set for task $(i, k)$ is $Z^{i,k} = \{Z_{j,l}^{i,k}\}_{j=1:n}^{l=0,1}$. The full data set for all training tasks is $Z^{\hat{S}} = (Z^{1,\hat{S}_1}, \ldots, Z^{\hat{n},\hat{S}_{\hat{n}}})$. The $j$th training sample for task pair $i$ is $Z_{j,S_j}^i = (Z_{j,S_j}^{i,0}, Z_{j,S_j}^{i,1})$. The $j$th training sample for all tasks is $Z_{j,S_j} = (Z_{j,S_j}^1, \ldots, Z_{j,S_j}^{\hat{n}})$. The training sets for all tasks is $Z_S = (Z_{1,S_1}, \ldots, Z_{n,S_n})$. The meta-training set is $Z_S^{\hat{S}} = (Z_{S^1}^{1,\hat{S}_1}, \ldots, Z_{S^{\hat{n}}}^{\hat{n},\hat{S}_{\hat{n}}})$. Finally, the output of the meta learner is $U$, the output of the base learner for task $(i, k)$ is $W^{i,k}$, and we let $W^i = (W^{i,0}, W^{i,1})$ and $W = (W^1, \ldots, W^{\hat{n}})$.

The conventions that we describe for $Z_{j,l}^{i,k}$ apply also for the losses $\lambda_{j,l}^{i,k}$, the instances $X_{j,l}^{i,k}$, the predictions $F_{j,l}^{i,k}$, and the representations $H_{j,l}^{i,k}$ that we consider in this appendix.

## A.1 Useful Lemmas

In this section, we present some lemmas that will be useful in the derivations of the main results. We begin with two change of measure inequalities for functions of random variables.

**Lemma 1** (Change of measure inequalities). *Let $X$ and $Y$ be two random variables over $\mathcal{X}$ and $\mathcal{Y}$ respectively, and let $Y'$ be a random variable with the same marginal distribution as $Y$ such that $Y'$ and $X$ are independent. Assume that the joint distribution of $X, Y$ is absolutely continuous with respect to the joint distribution of $X, Y'$. Let $f : \mathcal{X} \times \mathcal{Y} \to [-1, 1]$ and $g : \mathcal{X} \times \mathcal{Y} \to [0, 1]$ be measurable functions. Furthermore, assume that $\mathbb{E}_{X,Y'}[f(X, Y')] = 0$. Then, the following inequalities hold:*

$$|\mathbb{E}_{X,Y}[f(X, Y)]| \le \sqrt{2I(X; Y)}, \tag{26}$$

$$d(\mathbb{E}_{X,Y}[g(X, Y)] \,||\, \mathbb{E}_{X,Y'}[g(X, Y')]) \le I(X; Y). \tag{27}$$

*Proof.* Donsker-Varadhan's variational representation of the KL divergence implies that

$$\mathbb{E}_{X,Y}[\gamma f(X, Y)] \le I(X; Y) + \log \mathbb{E}_{X,Y'}\left[e^{\gamma f(X, Y')}\right]. \tag{28}$$

Now, $f(X, Y')$ is bounded to $[-1, 1]$ and $\mathbb{E}_{X,Y'}[f(X, Y')] = 0$. Therefore, $f(X, Y')$ is a sub-Gaussian random variable, which implies that

$$\log \mathbb{E}_{X,Y'}\left[e^{\gamma f(X, Y')}\right] \le \gamma^2/2. \tag{29}$$

Using this upper bound in (28), we obtain

$$\mathbb{E}_{X,Y}[\gamma f(X, Y)] - \gamma^2/2 \le I(X; Y). \tag{30}$$

By maximizing the left-hand side over $\gamma$, we establish (26).

We now turn to (27). Let $d_\gamma(q \,||\, p) = \gamma q - \log(1 - p + pe^\gamma)$, and note that this function is convex. By Jensen's inequality,

$$d_\gamma(\mathbb{E}_{X,Y}[g(X,Y)] \,||\, \mathbb{E}_{X,Y'}[g(X,Y')]) \leq \mathbb{E}_{X,Y}[d_\gamma(g(X,Y) \,||\, \mathbb{E}_{Y'}[g(X,Y')])]. \quad (31)$$

By Donsker-Varadhan's variational representation of the KL divergence,

$$\mathbb{E}_{X,Y}[d_\gamma(g(X,Y) \,||\, \mathbb{E}_{Y'}[g(X,Y')])] \leq I(X;Y) + \log \mathbb{E}_{X,Y'}\left[e^{d_\gamma\left(g(X,Y') \,||\, \mathbb{E}_{Y'}[g(X,Y')]\right)}\right]. \quad (32)$$

By [22, Eq. (17)], we have

$$\log \mathbb{E}_{X,Y'}\left[e^{d_\gamma\left(g(X,Y') \,||\, \mathbb{E}_{Y'}[g(X,Y')]\right)}\right] \leq 0. \quad (33)$$

Thus, by combining (31)-(33),

$$d_\gamma(\mathbb{E}_{X,Y}[g(X,Y)] \,||\, \mathbb{E}_{X,Y'}[g(X,Y')]) \leq I(X;Y). \quad (34)$$

The desired result follows because $\sup_\gamma d_\gamma(\cdot \,||\, \cdot) = d(\cdot \,||\, \cdot)$.

$\square$

**Lemma 2** (Conditioning on independent random variables). *Consider the random variables $X$, $Y$ and $Z$, where $X$ and $Z$ are independent. Then,*

$$I(X;Y) \leq I(X;Y|Z). \quad (35)$$

*Proof.* The result follows by using the independence of $X$ and $Z$ (which implies that $I(X;Z) = 0$), the chain rule for mutual information, and the non-negativity of mutual information as follows. Note that

$$I(X;Y,Z) = I(X;Z) + I(X;Y|Z) = I(X;Y|Z). \quad (36)$$

Alternatively,

$$I(X;Y,Z) = I(X;Y) + I(X;Z|Y) \geq I(X;Y). \quad (37)$$

Thus,

$$I(X;Y) \leq I(X;Y,Z) = I(X;Y|Z). \quad (38)$$

$\square$

**Lemma 3** (Full-sample relaxation). *Consider $n$ independent random variables $X = \{X_i\}_{i=1}^n$ and a random variable $Y$. Let $\phi$ be a convex function. Then,*

$$\frac{1}{n}\sum_{i=1}^n \phi(I(X_i;Y)) \leq \phi\left(\frac{I(X;Y)}{n}\right). \quad (39)$$

*Proof.* Let $X_{<i}$ denote $\{X_1, \ldots, X_{i-1}\}$. By the chain rule of mutual information,

$$I(X;Y) = \sum_{i=1}^n I(X_i;Y|X_{<i}). \quad (40)$$

Due to the independence of the $X_i$, Lemma 2 implies that $I(X_i;Y|X_{<i}) \geq I(X_i;Y)$. Combined with Jensen's inequality, this implies that

$$\phi\left(\frac{I(X;Y)}{n}\right) \geq \phi\left(\sum_{i=1}^n \frac{I(X_i;Y)}{n}\right) \geq \frac{1}{n}\sum_{i=1}^n \phi(I(X_i;Y)). \quad (41)$$

$\square$

**Lemma 4** (Sauer-Shelah lemma for the VC and Natarajan dimension). *Let $g_{\mathcal{F}}(\cdot)$ denote the growth function of the function class $\mathcal{F}$. Specifically, $g_{\mathcal{F}}(m)$ is the maximum number of different ways in which a data set of size $m$ can be classified using functions from $\mathcal{F}$. For any function class $\mathcal{F}$ with VC dimension $d_{\mathrm{VC}}$,*

$$g_{\mathcal{F}}(m) \leq \sum_{i=0}^{d_{\mathrm{VC}}} \binom{m}{i} \leq \begin{cases} 2^{d_{\mathrm{VC}}+1}, & m < d_{\mathrm{VC}} + 1 \\ \left(\dfrac{em}{d_{\mathrm{VC}}}\right)^{d_{\mathrm{VC}}}, & m \geq d_{\mathrm{VC}} + 1 \end{cases} \tag{42}$$

*More generally, for any function class $\mathcal{F}$ with range $\{0, \dots, N-1\}$ and Natarajan dimension $d_N$,*

$$g_{\mathcal{F}}(m) \leq \sum_{i=0}^{d_N} \binom{m}{i} \binom{N}{2}^i \leq \begin{cases} N^{d_N+1}, & m < d_N + 1, \\ \left(\binom{N}{2}\dfrac{em}{d_N}\right)^{d_N}, & m \geq d_N + 1. \end{cases} \tag{43}$$

*Proof.* The first inequality in (43) follows from [36, Cor. 5] and the second follows from [37, Lemma 10]. The result in (42) follows by setting $N = 2$ in (43), for which the Natarajan dimension $d_N$ coincides with the VC dimension [36, p. 222]. $\qquad\square$

## A.2 Proofs for Section 3.1

*Proof of Theorem 1.* We start by establishing a task-level generalization bound, i.e., a bound on $\left|L_{\mathcal{D}} - \widetilde{L}\right|$. By Jensen's inequality, the convexity of $|\cdot|$ implies that

$$\left|L_{\mathcal{D}} - \widetilde{L}\right| \leq \frac{1}{n\hat{n}} \sum_{i,j=1}^{\hat{n},n} \mathbb{E}_{Z,\hat{S}_i} \left[ \left| \mathbb{E}_{\lambda_j^{i,-\hat{S}_i}, S_j^{i,-\hat{S}_i}} \left[ \lambda_{j,-S_j^i}^{i,-\hat{S}_i} - \lambda_{j,S_j^i}^{i,-\hat{S}_i} \right] \right| \right]. \tag{44}$$

Let $S'$ be an independent copy of $S$. By symmetry, we see that

$$\mathbb{E}_{\lambda_j^{i,-\hat{S}_i}, S'_j{}^{i,-\hat{S}_i}} \left[ \lambda_{j,-S'_j{}^i}^{i,-\hat{S}_i} - \lambda_{j,S'_j{}^i}^{i,-\hat{S}_i} \right] = 0. \tag{45}$$

Using (26), we can therefore bound the argument of the expectation in (44) as

$$\left| \mathbb{E}_{\lambda_j^{i,-\hat{S}_i}, S_j^{i,-\hat{S}_i}} \left[ \lambda_{j,-S_j^i}^{i,-\hat{S}_i} - \lambda_{j,S_j^i}^{i,-\hat{S}_i} \right] \right| \leq \sqrt{2I^{Z,\hat{S}_i}(\lambda_j^{i,-\hat{S}_i}; S_j^{i,-\hat{S}_i})}. \tag{46}$$

Combining (44) and (46), we obtain the following task-level generalization bound:

$$\left|L_{\mathcal{D}} - \widetilde{L}\right| \leq \frac{1}{n\hat{n}} \sum_{i,j=1}^{\hat{n},n} \mathbb{E}_{Z,\hat{S}_i} \left[ \sqrt{2I^{Z,\hat{S}_i}(\lambda_j^{i,-\hat{S}_i}; S_j^{i,-\hat{S}_i})} \right]. \tag{47}$$

This is the first step of the two-step derivation.

Next, we establish an environment-level bound, i.e., a bound on $\left|\widetilde{L} - \widehat{L}\right|$. Again, by Jensen's inequality, the convexity of $|\cdot|$ implies that

$$\left|\widetilde{L} - \widehat{L}\right| \leq \frac{1}{n\hat{n}} \sum_{i,j=1}^{\hat{n},n} \mathbb{E}_{Z,S_j^i} \left[ \left| \mathbb{E}_{\lambda_{j,S_j^i}^i, \hat{S}_i} \left[ \lambda_{j,S_j^i}^{i,-\hat{S}_i} - \lambda_{j,S_j^i}^{i,\hat{S}_i} \right] \right| \right]. \tag{48}$$

Symmetry implies that

$$\mathbb{E}_{\lambda_{j,S_j^i}^i, \hat{S}'_i} \left[ \lambda_{j,S_j^i}^{i,-\hat{S}'_i} - \lambda_{j,S_j^i}^{i,\hat{S}'_i} \right] = 0. \tag{49}$$

We again bound the argument of the expectation, using (26), as

$$\left| \mathbb{E}_{\lambda_{j,S_j^i}^i, \hat{S}_i} \left[ \lambda_{j,S_j^i}^{i,-\hat{S}_i} - \lambda_{j,S_j^i}^{i,\hat{S}_i} \right] \right| \leq \sqrt{2I^{Z,S_j^i}(\lambda_{j,S_j^i}^i; \hat{S}_i)}. \tag{50}$$

Combining the two preceding inequalities, we obtain the following environment-level generalization guarantee, which is the second step:

$$\left| \widetilde{L} - \widehat{L} \right| \leq \frac{1}{n\hat{n}} \sum_{i,j=1}^{\hat{n},n} \mathbb{E}_{Z,S_j^i} \left[ \sqrt{2I^{Z,S_j^i}(\lambda_{j,S_j^i}^i; \hat{S}_i)} \right]. \tag{51}$$

We conclude the proof by observing that $\left| L_{\mathcal{D}} - \widehat{L} \right| \leq \left| L_{\mathcal{D}} - \widetilde{L} \right| + \left| \widetilde{L} - \widehat{L} \right|$ by the triangle inequality, and by using (47) and (51) to bound the two terms.

$\square$

*Proof of Corollary 1.* We begin with the first sum on the right-hand side of (4). By Jensen's inequality,

$$\frac{1}{n\hat{n}} \sum_{i,j=1}^{\hat{n},n} \mathbb{E}_{Z,S_j^i} \left[ \sqrt{2I^{Z,S_j^i}(\lambda_{j,S_j^i}^i; \hat{S}_i)} \right] \leq \sqrt{\frac{1}{n\hat{n}} \sum_{i,j=1}^{\hat{n},n} 2I(\lambda_{j,S_j^i}^i; \hat{S}_i | Z, S_j^i)}. \tag{52}$$

By Lemma 2 and the independence of $\hat{S}_i$ and $(S, R)$, we conclude that conditioning on $S$ and $R$ does not increase the mutual information. Hence,

$$\sqrt{\frac{1}{n\hat{n}} \sum_{i,j=1}^{\hat{n},n} 2I(\lambda_{j,S_j^i}^i; \hat{S}_i | Z, S_j^i)} \leq \sqrt{\frac{1}{n\hat{n}} \sum_{i,j=1}^{\hat{n},n} 2I(\lambda_{j,S_j^i}^i; \hat{S}_i | Z, S, R)}. \tag{53}$$

Since adding more random variables to the argument of the mutual information cannot decrease it, we have

$$\sqrt{\frac{1}{n\hat{n}} \sum_{i,j=1}^{\hat{n},n} 2I(\lambda_{j,S_j^i}^i; \hat{S}_i | Z, S, R)} \leq \sqrt{\frac{1}{n\hat{n}} \sum_{i,j=1}^{\hat{n},n} 2I(\lambda_{j,S_j^i}; \hat{S}_i | Z, S, R)}, \tag{54}$$

where $\lambda_{j,S_j^i} = \{\lambda_{j,S_j^i}^i\}_{i=1}^{\hat{n}}$. Since the $\hat{S}_i$ are independent, it follows from Lemma 3 that

$$\sqrt{\frac{1}{n\hat{n}} \sum_{i,j=1}^{\hat{n},n} 2I(\lambda_{j,S_j^i}; \hat{S}_i | Z, S, R)} \leq \sqrt{\frac{1}{n} \sum_{j=1}^{n} \frac{2I(\lambda_{j,S_j^i}; \hat{S} | Z, S, R)}{\hat{n}}}. \tag{55}$$

Now, note that given $Z$, $S$ and $R$, the losses $\lambda_{j,S_j^i}$ are a function of the output of the meta learner $U$. Therefore,

$$\sqrt{\frac{1}{n} \sum_{j=1}^{n} \frac{2I(\lambda_{j,S_j^i}; \hat{S} | Z, S, R)}{\hat{n}}} \leq \sqrt{\frac{2I(U; \hat{S} | Z, S, R)}{\hat{n}}} \leq \sqrt{\frac{2I(U; \hat{S} | Z, S)}{\hat{n}}}, \tag{56}$$

where the last step follows from the independence of $U$ and $R$. By combining (52)-(56), we can bound the first sum in the right-hand side of (4) as

$$\frac{1}{n\hat{n}} \sum_{i,j=1}^{\hat{n},n} \mathbb{E}_{Z,S_j^i} \left[ \sqrt{2I^{Z,S_j^i}(\lambda_{j,S_j^i}^i; \hat{S}_i)} \right] \leq \sqrt{\frac{2I(U; \hat{S} | Z, S)}{\hat{n}}}. \tag{57}$$

For the second sum on the right-hand side of (4), we again use Jensen's inequality to conclude that

$$\frac{1}{n\hat{n}} \sum_{i,j=1}^{\hat{n},n} \mathbb{E}_{Z,\hat{S}_i} \left[ \sqrt{2I^{Z,\hat{S}_i}(\lambda_j^{i,-\hat{S}_i}; S_j^{i,-\hat{S}_i})} \right] \leq \sqrt{\frac{1}{n\hat{n}} \sum_{i,j=1}^{\hat{n},n} 2I(\lambda_j^{i,-\hat{S}_i}; S_j^{i,-\hat{S}_i} | Z, \hat{S}_i)}. \tag{58}$$

Since adding more random variables does not decrease the mutual information,

$$\sqrt{\frac{1}{n\hat{n}}\sum_{i,j=1}^{\hat{n},n}2I(\lambda_j^{i,-\hat{S}_i};S_j^{i,-\hat{S}_i}|Z,\hat{S}_i)} \leq \sqrt{\frac{1}{n\hat{n}}\sum_{i,j=1}^{\hat{n},n}2I(\lambda^{i,-\hat{S}_i};S_j^{i,-\hat{S}_i}|Z,\hat{S}_i)}. \qquad (59)$$

By Lemma 2 and the independence of the $S_j^{i,-\hat{S}_i}$,

$$\sqrt{\frac{1}{n\hat{n}}\sum_{i,j=1}^{\hat{n},n}2I(\lambda^{i,-\hat{S}_i};S_j^{i,-\hat{S}_i}|Z,\hat{S}_i)} \leq \sqrt{\frac{1}{\hat{n}}\sum_{i=1}^{\hat{n}}\frac{2I(\lambda^{i,-\hat{S}_i};S^{i,-\hat{S}_i}|Z,\hat{S}_i)}{n}}. \qquad (60)$$

Since $R_i$, $\hat{S}_i$ and $S^{i,-\hat{S}_i}$ have the same distribution for all $i=1,\ldots,\hat{n}$,

$$\sqrt{\frac{1}{\hat{n}}\sum_{i=1}^{\hat{n}}\frac{2I(\lambda^{i,-\hat{S}_i};S^{i,-\hat{S}_i}|Z,\hat{S}_i)}{n}} = \sqrt{\frac{2I(\lambda^{1,-\hat{S}_1};S^{1,-\hat{S}_1}|Z,\hat{S}_i)}{n}}. \qquad (61)$$

Finally, given $Z$ and $\hat{S}_i$, $\lambda^{1,-\hat{S}_1}$ is a function of $W^{1,-\hat{S}_1}$. Hence,

$$\sqrt{\frac{2I(\lambda^{1,-\hat{S}_1};S^{1,-\hat{S}_1}|Z,\hat{S}_i)}{n}} \leq \sqrt{\frac{2I(W^{1,-\hat{S}_1};S^{1,-\hat{S}_1}|Z,\hat{S}_i)}{n}}. \qquad (62)$$

By combining (58)-(62), we can bound the second term in the right-hand side of (4) as

$$\frac{1}{n\hat{n}}\sum_{i,j=1}^{\hat{n},n}\mathbb{E}_{Z,\hat{S}_i}\left[\sqrt{2I^{Z,\hat{S}_i}(\lambda_j^{i,-\hat{S}_i};S_j^{i,-\hat{S}_i})}\right] \leq \sqrt{\frac{2I(W^{1,-\hat{S}_1};S^{1,-\hat{S}_1}|Z,\hat{S}_i)}{n}}. \qquad (63)$$

The result follows by combining (57) and (63). $\qquad\square$

*Proof of Theorem 2.* By Jensen's inequality, we have

$$\left|L_{\mathcal{D}}-\widehat{L}\right| \leq \frac{1}{n\hat{n}}\sum_{i,j=1}^{\hat{n},n}\mathbb{E}_Z\left[\left|\mathbb{E}_{\lambda_j^i,\hat{S}_i,S_j^i}\left[\lambda_{j,-S_j^i}^{i,-\hat{S}_i}-\lambda_{j,S_j^i}^{i,\hat{S}_i}\right]\right|\right]. \qquad (64)$$

Now, let $\hat{S}'$ and $S'$ be independent copies of $\hat{S}$ and $S$. Note that

$$\mathbb{E}_{\lambda_j^i,\hat{S}'_i,S_j'^{i}}\left[\lambda_{j,-S_j'^{i}}^{i,-\hat{S}'_i}-\lambda_{j,S_j'^{i}}^{i,\hat{S}'_i}\right]=0. \qquad (65)$$

We can therefore apply (26), with $X=\lambda_j^i$ and $Y$ being the pair of random variables $(\hat{S}_i,S_j'^{i})$, to bound the argument of the expectation as

$$\left|\mathbb{E}_{\lambda_j^i,\hat{S}_i,S_j^i}\left[\lambda_{j,-S_j^i}^{i,-\hat{S}_i}-\lambda_{j,S_j^i}^{i,\hat{S}_i}\right]\right| \leq \sqrt{2I^Z(\lambda_j^i;\hat{S}_i,S_j^i)}. \qquad (66)$$

Combining (64) and (66), we establish the desired result.

$$\square$$

*Proof of Corollary 2.* By Jensen's inequality,

$$\frac{1}{n\hat{n}}\sum_{i,j=1}^{\hat{n},n}\mathbb{E}_Z\left[\sqrt{2I^Z(\lambda_j^i;\hat{S}_i,S_j^i)}\right] \leq \sqrt{\frac{1}{n\hat{n}}\sum_{i,j=1}^{\hat{n},n}2I(\lambda_j^i;\hat{S}_i,S_j^i|Z)}. \qquad (67)$$

Since adding more random variables does not decrease the mutual information,

$$\sqrt{\frac{1}{n\hat{n}}\sum_{i,j=1}^{\hat{n},n} 2I(\lambda_j^i; \hat{S}_i, S_j^i|Z)} \leq \sqrt{\frac{1}{n\hat{n}}\sum_{i,j=1}^{\hat{n},n} 2I(\lambda; \hat{S}_i, S_j^i|Z)}, \tag{68}$$

where $\lambda = \{\lambda_j^i\}_{j=1:n}^{i=1:\hat{n}}$. By the independence of the $S_j^i$ for different $j$, and $(\hat{S}_i, S^i)$ for different $i$,

$$\sqrt{\frac{1}{n\hat{n}}\sum_{i,j=1}^{\hat{n},n} 2I(\lambda; \hat{S}_i, S_j^i|Z)} \leq \sqrt{\frac{2I(\lambda; \hat{S}, S|Z)}{n\hat{n}}}. \tag{69}$$

Given $Z$, the losses $\lambda$ are a function of $W = \{W^i\}_{i=1}^{\hat{n}}$. Thus,

$$\sqrt{\frac{2I(\lambda; \hat{S}, S|Z)}{n\hat{n}}} \leq \sqrt{\frac{2I(W; \hat{S}, S|Z)}{n\hat{n}}}. \tag{70}$$

Combining (67)-(70), we establish the first inequality in (7). Next, since adding random variables does not decrease mutual information,

$$\sqrt{\frac{2I(W; \hat{S}, S|Z)}{n\hat{n}}} \leq \sqrt{\frac{2I(U, W; \hat{S}, S|Z)}{n\hat{n}}} \tag{71}$$

$$\leq \sqrt{\frac{2I(U; \hat{S}, S|Z) + 2I(W; \hat{S}, S|Z, U)}{n\hat{n}}} \tag{72}$$

where the second step follows from the chain rule. Since the conditional distribution $P_{W\hat{S}S|ZU}$ factorizes as $P_{\hat{S}|ZU}\prod_{i=1}^{\hat{n}} P_{W^i S^i|ZU}$, we have that

$$I(W; \hat{S}, S|Z, U) = \sum_{i=1}^{\hat{n}} I(W^i; S^i|Z, U). \tag{73}$$

Furthermore, since $(R_i, \hat{S}_i, S^i)$ are identically distributed for all $i$,

$$\sum_{i=1}^{\hat{n}} I(W^i; S^i|Z, U) = \hat{n}I(W^1; S^1|Z, U). \tag{74}$$

By combining (70)-(74), we get

$$\sqrt{\frac{2I(W; \hat{S}, S|Z)}{n\hat{n}}} \leq \sqrt{\frac{2I(U; \hat{S}, S|Z) + 2\hat{n}I(W^1; S^1|Z, U)}{n\hat{n}}}. \tag{75}$$

This establishes the second inequality in (7).

Finally, by the chain rule,

$$I(U; \hat{S}, S|Z) \leq I(U; \hat{S}, S|Z) + I(U; Z) = I(U; Z, \hat{S}, S) = I(U; Z_S^{\hat{S}}). \tag{76}$$

Similarly,

$$I(W^1; S^1|Z, U) \leq I(W^1; S^1|Z, U) + I(W^1; Z|U) = I(W^1; S^1, Z|U) = I(W^1; Z_{S^1}^1|U). \tag{77}$$

By combining (75)-(77), we establish (8). Thus, to summarize, we have shown that

$$\left| L_{\mathcal{D}} - \widehat{L} \right| \leq \sqrt{\frac{2I(W; \hat{S}, S|Z)}{n\hat{n}}} \leq \sqrt{\frac{2I(U; \hat{S}, S|Z) + 2\hat{n}I(W^1; S^1|Z, U)}{n\hat{n}}} \tag{78}$$

$$\leq \sqrt{\frac{2I(U; Z_S^{\hat{S}}) + 2\hat{n}I(W^1; Z_{S^1}^1|U)}{n\hat{n}}}. \tag{79}$$

$\square$

*Proof of Theorem 3.* We begin by proving (10). First, we derive a task-level generalization bound. By Jensen's inequality, we have

$$d\left( \widetilde{L} \,\|\, \frac{\widetilde{L} + L_{\mathcal{D}}}{2} \right) \leq \frac{1}{n\hat{n}} \sum_{i,j=1}^{\hat{n},n} \mathbb{E}_{Z,\hat{S}_i} \left[ \mathbb{E}_{\lambda_j^{i,-\hat{S}_i}, S_j^{i,-\hat{S}_i}} \left[ d\left( \lambda_{j,S_j^i}^{i,-\hat{S}_i} \,\|\, \frac{\lambda_{j,-S_j^i}^{i,-\hat{S}_i} + \lambda_{j,S_j^i}^{i,-\hat{S}_i}}{2} \right) \right] \right]. \tag{80}$$

Since $S_j^{i,-\hat{S}_i} \in \{0,1\}$, $\lambda_{j,-S_j^i}^{i,-\hat{S}_i} + \lambda_{j,S_j^i}^{i,-\hat{S}_i} = \lambda_{j,0}^{i,-\hat{S}_i} + \lambda_{j,1}^{i,-\hat{S}_i}$ does not actually depend on $S_j^{i,-\hat{S}_i}$. Now, let $S'$ be an independent copy of $S$. It follows that

$$\mathbb{E}_{\lambda_j^{i,-\hat{S}_i}, S_j'^{i,-\hat{S}_i}} \left[ \lambda_{j,S_j'^i}^{i,-\hat{S}_i} \right] = \mathbb{E}_{\lambda_j^{i,-\hat{S}_i}} \left[ \frac{\lambda_{j,0}^{i,-\hat{S}_i} + \lambda_{j,1}^{i,-\hat{S}_i}}{2} \right]. \tag{81}$$

We can thus use (26) to bound the argument of the expectation as

$$\mathbb{E}_{\lambda_j^{i,-\hat{S}_i}, S_j^{i,-\hat{S}_i}} \left[ d\left( \lambda_{j,S_j^i}^{i,-\hat{S}_i} \,\|\, \frac{\lambda_{j,-S_j^i}^{i,-\hat{S}_i} + \lambda_{j,S_j^i}^{i,-\hat{S}_i}}{2} \right) \right] \leq I^{Z,\hat{S}_i}(\lambda_j^{i,-\hat{S}_i}; S_j^{i,-\hat{S}_i}). \tag{82}$$

Combining the two inequalities, we obtain

$$d\left( \widetilde{L} \,\|\, \frac{\widetilde{L} + L_{\mathcal{D}}}{2} \right) \leq \frac{1}{n\hat{n}} \sum_{i,j=1}^{\hat{n},n} I(\lambda_j^{i,-\hat{S}_i}; S_j^{i,-\hat{S}_i}|Z, \hat{S}_i). \tag{83}$$

Recall that

$$d_2^{-1}(q, c) = \sup \left\{ p \in [0,1] : d\left( q \,\|\, \frac{q+p}{2} \right) \leq c \right\}. \tag{84}$$

Using $d_2^{-1}(\cdot)$ to invert (83), we get

$$L_{\mathcal{D}} \leq d_2^{-1}\left( \widetilde{L}, \frac{1}{n\hat{n}} \sum_{i,j=1}^{\hat{n},n} I(\lambda_j^{i,-\hat{S}_i}; S_j^{i,-\hat{S}_i}|Z, \hat{S}_i) \right). \tag{85}$$

Next, we perform similar steps at the environment level. First, by Jensen's inequality,

$$d\left( \widehat{L} \,\|\, \frac{\widehat{L} + \widetilde{L}}{2} \right) \leq \frac{1}{n\hat{n}} \sum_{i,j=1}^{\hat{n},n} \mathbb{E}_{Z,S^i} \left[ \mathbb{E}_{\lambda_{j,S_j^i}^i, \hat{S}_i} \left[ d\left( \lambda_{j,S_j^i}^{i,\hat{S}_i} \,\|\, \frac{\lambda_{j,S_j^i}^{i,\hat{S}_i} + \lambda_{j,S_j^i}^{i,-\hat{S}_i}}{2} \right) \right] \right]. \tag{86}$$

Let $\hat{S}'$ be an independent copy of $\hat{S}$. By a similar argument as in the proof of the task-level bound,

$$\mathbb{E}_{\lambda_{j,S_j^i}^i, \hat{S}_i'} \left[ \lambda_{j,S_j^i}^{i,\hat{S}_i'} \right] = \mathbb{E}_{\lambda_{j,S_j^i}^i} \left[ \frac{\lambda_{j,S_j^i}^{i,0} + \lambda_{j,S_j^i}^{i,1}}{2} \right]. \tag{87}$$

We can therefore again bound the argument of the expectation with (27) to obtain

$$\mathbb{E}_{\lambda^i_{j,S^i_j},\hat{S}_i}\left[d\left(\lambda^{i,\hat{S}_i}_{j,S^i_j}\,||\,\frac{\lambda^{i,\hat{S}_i}_{j,S^i_j}+\lambda^{i,-\hat{S}_i}_{j,S^i_j}}{2}\right)\right]\leq I^{Z,S^i_j}(\lambda^i_{j,S^i_j};\hat{S}_i). \tag{88}$$

By combining the two inequalities, we find that

$$d\left(\widehat{L}\,||\,\frac{\widehat{L}+\widetilde{L}}{2}\right)\leq\frac{1}{n\hat{n}}\sum_{i,j=1}^{\hat{n},n}I(\lambda^i_{j,S^i_j};\hat{S}_i|Z,S^i_j) \tag{89}$$

which, through the use of $d_2^{-1}(\cdot)$, implies that

$$\widetilde{L}\leq d_2^{-1}\left(\widehat{L},\frac{1}{n\hat{n}}\sum_{i,j=1}^{\hat{n},n}I(\lambda^i_{j,S^i_j};\hat{S}_i|Z,S^i_j)\right). \tag{90}$$

To complete the proof, we use the following observation. Assume that $L_{\mathcal{D}}\leq B(\widetilde{L})$, where $B(\cdot)$ is a non-decreasing function. Then, if $\widetilde{L}\leq\widehat{B}(\widehat{L})$, we have $L_{\mathcal{D}}\leq B(\widehat{B}(\widehat{L}))$. To apply this observation, we note that $d_2^{-1}(\cdot,c)$ is non-decreasing for $c>0$. Chaining the two bounds, we obtain

$$L_{\mathcal{D}}\leq d_2^{-1}\left(\widetilde{L},\frac{1}{n\hat{n}}\sum_{i,j=1}^{\hat{n},n}I(\lambda^{i,-\hat{S}_i}_j;S^{i,-\hat{S}_i}_j|Z,\hat{S}_i)\right) \tag{91}$$

$$\leq d_2^{-1}\left(d_2^{-1}\left(\widehat{L},\frac{1}{n\hat{n}}\sum_{i,j=1}^{\hat{n},n}I(\lambda^i_{j,S^i_j};\hat{S}_i|Z,S^i_j)\right),\frac{1}{n\hat{n}}\sum_{i,j=1}^{\hat{n},n}I(\lambda^{i,-\hat{S}_i}_j;S^{i,-\hat{S}_i}_j|Z,\hat{S}_i)\right). \tag{92}$$

This establishes (10).

Next, we turn to (11). By Jensen's inequality, we have

$$d\left(\widehat{L}\,||\,\frac{\widehat{L}+\bar{L}+\widetilde{L}+L_{\mathcal{D}}}{4}\right)$$

$$\leq\frac{1}{n\hat{n}}\sum_{i,j=1}^{\hat{n},n}\mathbb{E}_Z\left[\mathbb{E}_{\lambda^i_j,\hat{S}_i,S^i_j}\left[d\left(\lambda^{i,\hat{S}_i}_{j,S^i_j}\,||\,\frac{\lambda^{i,\hat{S}_i}_{j,S^i_j}+\lambda^{i,\hat{S}_i}_{j,-S^i_j}+\lambda^{i,-\hat{S}_i}_{j,S^i_j}+\lambda^{i,-\hat{S}_i}_{j,-S^i_j}}{4}\right)\right]\right]. \tag{93}$$

Let $\hat{S}'$ and $S'$ be independent copies of $\hat{S}$ and $S$ respectively. We note that

$$\mathbb{E}_{\lambda^i_j,\hat{S}'_i,S'^i_j}\left[\lambda^{i,\hat{S}_i}_{j,S^i_j}\right]=\mathbb{E}_{\lambda^i_j}\left[\frac{\lambda^{i,0}_{j,0}+\lambda^{i,0}_{j,1}+\lambda^{i,1}_{j,0}+\lambda^{i,1}_{j,1}}{4}\right]. \tag{94}$$

This means that we can apply (27) to the argument of the expectation to get

$$\mathbb{E}_{\lambda^i_j,\hat{S}_i,S^i_j}\left[d\left(\lambda^{i,\hat{S}_i}_{j,S^i_j}\,||\,\frac{\lambda^{i,\hat{S}_i}_{j,S^i_j}+\lambda^{i,\hat{S}_i}_{j,-S^i_j}+\lambda^{i,-\hat{S}_i}_{j,S^i_j}+\lambda^{i,-\hat{S}_i}_{j,-S^i_j}}{4}\right)\right]\leq I^Z(\lambda^i_j;\hat{S}_i,S^i_j). \tag{95}$$

The result in (11) now follows by combining (93) and (95).

$\square$

*Proof of Corollary 3.* Since $\lim_{x \to 0^+} x \log x = 0$, we use the convention that $0 \log 0 = 0$. From the definition of $d(\cdot \,||\, \cdot)$, we thus get

$$d\left(0 \,||\, \frac{p}{4}\right) = \log \frac{1}{1 - \frac{p}{4}} \tag{96}$$

from which it follows that

$$d_4^{-1}(0, c) = 4 - 4e^{-c}. \tag{97}$$

By the non-negativity of the loss function, $L_{\mathcal{D}} \leq L_{\mathcal{D}} + \bar{L} + \widetilde{L}$. Thus, combining (11) with (97), we obtain the result in (12).

$\square$

## A.3 Proofs for Section 3.2

To derive the simplified result stated in Theorem 4, we assume that the meta learner and base learner are invariant to the order of the data samples. However, this assumption is only necessary to simplify the expression, and a similar bound holds more generally without this assumption. Therefore, we first state and prove this more general result in Theorem 6. Then, we describe how to simplify the result to obtain Theorem 4. Later, when proving Corollary 6, we will use the more general Theorem 6 as the basis of the derivation.

**Theorem 6.** *Consider the setting introduced in Section 2. For each $j$, let $Q_{j,S_j}$ denote the conditional distribution of $\lambda_{j,S_j}$ given $(Z, \hat{S}, S)$, and let $P_{j,S_j}$ denote $\mathbb{E}_{\hat{S}}\left[Q_{j,S_j}\right]$. Furthermore, let $Q^{i,-\hat{S}_i}$ denote the conditional distribution of $\lambda^{i,-\hat{S}_i}$ given $(Z, \hat{S}, S)$, and let $P^{i,-\hat{S}_i}$ denote $\mathbb{E}_{S-\hat{s}}\left[Q^{i,-\hat{S}_i}\right]$. Then, with probability at least $1 - \delta$ under the draw of $(Z, \hat{S}, S)$,*

$$\left| L_{\mathcal{D}} - \widehat{L} \right| \leq \sqrt{\frac{\frac{1}{n}\sum_{j=1}^{n} 2D(Q_{j,S_j} \,||\, P_{j,S_j}) + 2\log(\frac{2n\sqrt{\hat{n}}}{\delta})}{\hat{n} - 1}}$$
$$+ \sqrt{\frac{\sum_{i=1}^{\hat{n}} 2D(Q^{i,-\hat{S}_i} \,||\, P^{i,-\hat{S}_i}) + 2\log(\frac{2\sqrt{n\hat{n}}}{\delta})}{n\hat{n} - 1}}. \tag{98}$$

*Proof of Theorem 6.* First, let $\widetilde{L}(Z, \hat{S}, S)$ denote the training loss on unobserved tasks,

$$\widetilde{L}(Z, \hat{S}, S) = \frac{1}{n\hat{n}} \sum_{i,j=1}^{\hat{n},n} \mathbb{E}_{\hat{R}, R_i}\left[\ell(\mathcal{A}(Z_{S^i}^{i,-\hat{S}_i}, R_i, \hat{\mathcal{A}}(Z_{\hat{S}}^{\hat{S}}, \hat{R})), Z_{j,-S_j^i}^{i,-\hat{S}_i})\right]. \tag{99}$$

We begin by establishing an environment-level bound. Let $\lambda_{j,S_j}$ be distributed according to $Q_{j,S_j}$. By Jensen's inequality,

$$\frac{\hat{n}-1}{2}\left(\widetilde{L}(Z,\hat{S},S) - \widehat{L}(Z,\hat{S},S)\right)^2 \leq \frac{1}{n}\sum_{j=1}^{n}\mathbb{E}_{\lambda_{j,S_j}}\left[\frac{\hat{n}-1}{2}\left(\frac{1}{\hat{n}}\sum_{i=1}^{\hat{n}}\lambda_{j,S_j}^{i,-\hat{S}_i} - \lambda_{j,S_j^i}^{i,\hat{S}_i}\right)^2\right]. \tag{100}$$

Now, let $\lambda'_{j,S_j}$ be distributed according to $P_{j,S_j}$. By Donsker-Varadhan's variational representation of the KL divergence,

$$\frac{1}{n}\sum_{j=1}^{n}\mathbb{E}_{\lambda_{j,S_j}}\left[\frac{\hat{n}-1}{2}\left(\frac{1}{\hat{n}}\sum_{i=1}^{\hat{n}}\lambda_{j,S_j}^{i,-\hat{S}_i} - \lambda_{j,S_j^i}^{i,\hat{S}_i}\right)^2\right]$$
$$\leq \frac{1}{n}\sum_{j=1}^{n}\left(D(Q_{j,S_j} \,||\, P_{j,S_j}) + \log\mathbb{E}_{\lambda'_{j,S_j}}\left[\exp\left(\frac{\hat{n}-1}{2}\left(\frac{1}{\hat{n}}\sum_{i=1}^{\hat{n}}\lambda'^{i,-\hat{S}_i}_{j,S_j} - \lambda'^{i,\hat{S}_i}_{j,S_j}\right)^2\right)\right]\right). \tag{101}$$

For each $j$, Markov's inequality implies that, with probability at least $1-\delta$ under the draw of $(Z, \hat{S}, S)$,

$$D(Q_{j,S_j} \,\|\, P_{j,S_j}) + \log \mathbb{E}_{\lambda'_{j,S_j}}\left[\exp\left(\frac{\hat{n}-1}{2}\left(\frac{1}{\hat{n}}\sum_{i=1}^{\hat{n}} \lambda'^{i,-\hat{S}_i}_{j,S_j^i} - \lambda'^{i,\hat{S}_i}_{j,S_j^i}\right)^2\right)\right]$$

$$\leq D(Q_{j,S_j} \,\|\, P_{j,S_j}) + \log \mathbb{E}_{\lambda'_{j,S_j},Z,\hat{S},S}\left[\frac{1}{\delta}\exp\left(\frac{\hat{n}-1}{2}\left(\frac{1}{\hat{n}}\sum_{i=1}^{\hat{n}} \lambda'^{i,-\hat{S}_i}_{j,S_j^i} - \lambda'^{i,\hat{S}_i}_{j,S_j^i}\right)^2\right)\right]. \quad (102)$$

By the union bound, this implies that, with $\delta \to \delta/n$, (102) holds for all $j$ simultaneously with probability at least $1 - \delta$. Thus, with probability at least $1 - \delta$ under the draw of $(Z, \hat{S}, S)$,

$$\frac{1}{n}\sum_{j=1}^{n}\left(D(Q_{j,S_j} \,\|\, P_{j,S_j}) + \log \mathbb{E}_{\lambda'_{j,S_j}}\left[\exp\left(\frac{\hat{n}-1}{2}\left(\frac{1}{\hat{n}}\sum_{i=1}^{\hat{n}} \lambda'^{i,-\hat{S}_i}_{j,S_j^i} - \lambda'^{i,\hat{S}_i}_{j,S_j^i}\right)^2\right)\right]\right)$$

$$\leq \frac{1}{n}\sum_{j=1}^{n}\left(D(Q_{j,S_j}\|P_{j,S_j}) + \log \mathbb{E}_{\lambda'_{j,S_j},Z,\hat{S},S}\left[\frac{n}{\delta}\exp\left(\frac{\hat{n}-1}{2}\left(\frac{1}{\hat{n}}\sum_{i=1}^{\hat{n}} \lambda'^{i,-\hat{S}_i}_{j,S_j^i} - \lambda'^{i,\hat{S}_i}_{j,S_j^i}\right)^2\right)\right]\right). \quad (103)$$

Note that, on the right-hand side of (103), $\hat{S}$ is independent from $(\lambda'_{j,S_j}, Z, S)$. Furthermore, for each $(i,j)$, $\lambda'^{i,-\hat{S}_i}_{j,S_j^i} - \lambda'^{i,\hat{S}_i}_{j,S_j^i}$ is bounded to $[-1,1]$ and $\mathbb{E}_{\hat{S}_i}\left[\lambda'^{i,-\hat{S}_i}_{j,S_j^i} - \lambda'^{i,\hat{S}_i}_{j,S_j^i}\right] = 0$. This implies that $\frac{1}{\hat{n}}\sum_{i=1}^{\hat{n}} \lambda'^{i,-\hat{S}_i}_{j,S_j^i} - \lambda'^{i,\hat{S}_i}_{j,S_j^i}$ is a $1/\sqrt{\hat{n}}$-sub-Gaussian random variable, from which it follows that [38, Thm. 2.6.(IV)]

$$\log \mathbb{E}_{\lambda'_{j,S_j},Z,\hat{S},S}\left[\exp\left(\frac{\hat{n}-1}{2}\left(\frac{1}{\hat{n}}\sum_{i=1}^{\hat{n}} \lambda'^{i,-\hat{S}_i}_{j,S_j^i} - \lambda'^{i,\hat{S}_i}_{j,S_j^i}\right)^2\right)\right] \leq \log(\sqrt{\hat{n}}). \quad (104)$$

By substituting (104) into (103), we obtain

$$\frac{1}{n}\sum_{j=1}^{n}\left(D(Q_{j,S_j} \,\|\, P_{j,S_j}) + \log \mathbb{E}_{\lambda'_{j,S_j}}\left[\exp\left(\frac{\hat{n}-1}{2}\left(\frac{1}{\hat{n}}\sum_{i=1}^{\hat{n}} \lambda'^{i,-\hat{S}_i}_{j,S_j^i} - \lambda'^{i,\hat{S}_i}_{j,S_j^i}\right)^2\right)\right]\right)$$

$$\leq \frac{1}{n}\sum_{j=1}^{n} D(Q_{j,S_j} \,\|\, P_{j,S_j}) + \log\left(\frac{n\sqrt{\hat{n}}}{\delta}\right). \quad (105)$$

By combining (100)-(105), we get, after some arithmetic,

$$\left|\widetilde{L}(Z, \hat{S}, S) - \widehat{L}(Z, \hat{S}, S)\right| \leq \sqrt{\frac{\frac{1}{n}\sum_{j=1}^{n} 2D(Q_{j,S_j} \,\|\, P_{j,S_j}) + 2\log\left(\frac{n\sqrt{\hat{n}}}{\delta}\right)}{\hat{n}-1}}. \quad (106)$$

We now turn to the task level. Let $Q^{-\hat{S}}$ denote the conditional distribution of $\lambda^{-\hat{S}}$ given $(Z, \hat{S}, S)$, and let $P^{-\hat{S}}$ denote $\mathbb{E}_{S^{-\hat{S}}}\left[Q^{-\hat{S}}\right]$. Let $\lambda^{-\hat{S}}$ be distributed according to $Q^{-\hat{S}}$. By Jensen's inequality,

$$\frac{n\hat{n}-1}{2}\left(L_{\mathcal{D}}(Z, \hat{S}, S) - \widetilde{L}(Z, \hat{S}, S)\right)^2 \leq \mathbb{E}_{\lambda^{-\hat{S}}}\left[\frac{n\hat{n}-1}{2}\left(\frac{1}{n\hat{n}}\sum_{i,j=1}^{\hat{n},n} \lambda^{i,-\hat{S}_i}_{j,-S_j^i} - \lambda^{i,-\hat{S}_i}_{j,S_j^i}\right)^2\right]. \quad (107)$$

Now, let $\lambda'^{-\hat{S}}$ be distributed according to $P^{-\hat{S}}$. By Donsker-Varadhan's variational representation of the KL divergence,

$$
\mathbb{E}_{\lambda^{-\hat{S}}} \left[ \frac{n\hat{n}-1}{2} \left( \frac{1}{n\hat{n}} \sum_{i,j=1}^{\hat{n},n} \lambda_{j,-S_j^i}^{i,-\hat{S}_i} - \lambda_{j,S_j^i}^{i,-\hat{S}_i} \right)^2 \right]
$$
$$
\leq D(Q^{-\hat{S}} \,\|\, P^{-\hat{S}}) + \log \mathbb{E}_{\lambda'^{-\hat{S}}} \left[ \exp \left( \frac{n\hat{n}-1}{2} \left( \frac{1}{n\hat{n}} \sum_{i,j=1}^{\hat{n},n} \lambda'^{i,-\hat{S}_i}_{j,-S_j^i} - \lambda'^{i,-\hat{S}_i}_{j,S_j^i} \right)^2 \right) \right]. \quad (108)
$$

By Markov's inequality, we conclude that with probability at least $1 - \delta$ under $(Z, \hat{S}, S)$,

$$
D(Q^{-\hat{S}} \,\|\, P^{-\hat{S}}) + \log \mathbb{E}_{\lambda'^{-\hat{S}}} \left[ \exp \left( \frac{n\hat{n}-1}{2} \left( \frac{1}{n\hat{n}} \sum_{i,j=1}^{\hat{n},n} \lambda'^{i,-\hat{S}_i}_{j,-S_j^i} - \lambda'^{i,-\hat{S}_i}_{j,S_j^i} \right)^2 \right) \right]
$$
$$
\leq D(Q^{-\hat{S}} \,\|\, P^{-\hat{S}}) + \log \mathbb{E}_{\lambda'^{-\hat{S}},Z,\hat{S},S} \left[ \frac{1}{\delta} \exp \left( \frac{n\hat{n}-1}{2} \left( \frac{1}{n\hat{n}} \sum_{i,j=1}^{\hat{n},n} \lambda'^{i,-\hat{S}_i}_{j,-S_j^i} - \lambda'^{i,-\hat{S}_i}_{j,S_j^i} \right)^2 \right) \right]. \quad (109)
$$

Note that $S^{-\hat{S}}$ is independent from $(\lambda'^{-\hat{S}}, Z, \hat{S}, S^{\hat{S}})$, that for each $(i,j)$, $\lambda'^{i,-\hat{S}_i}_{j,-S_j^i} - \lambda'^{i,-\hat{S}_i}_{j,S_j^i}$ is bounded to $[-1,1]$, and that $\mathbb{E}_{S_j^{i,-\hat{S}_i}} \left[ \lambda'^{i,-\hat{S}_i}_{j,-S_j^i} - \lambda'^{i,-\hat{S}_i}_{j,S_j^i} \right] = 0$. Thus, it follows that [38, Thm. 2.6.(IV)]

$$
\log \mathbb{E}_{\lambda'^{-\hat{S}},Z,\hat{S},S} \left[ \exp \left( \frac{n\hat{n}-1}{2} \left( \frac{1}{n\hat{n}} \sum_{i,j=1}^{\hat{n},n} \lambda'^{i,-\hat{S}_i}_{j,-S_j^i} - \lambda'^{i,-\hat{S}_i}_{j,S_j^i} \right)^2 \right) \right] \leq \log \left( \sqrt{n\hat{n}} \right). \quad (110)
$$

By substituting (110) into (109), we obtain

$$
D(Q^{-\hat{S}} \,\|\, P^{-\hat{S}}) + \log \mathbb{E}_{\lambda'^{-\hat{S}}} \left[ \exp \left( \frac{n\hat{n}-1}{2} \left( \frac{1}{n\hat{n}} \sum_{i,j=1}^{\hat{n},n} \lambda'^{i,-\hat{S}_i}_{j,-S_j^i} - \lambda'^{i,-\hat{S}_i}_{j,S_j^i} \right)^2 \right) \right]
$$
$$
\leq D(Q^{-\hat{S}} \,\|\, P^{-\hat{S}}) + \log \left( \frac{\sqrt{n\hat{n}}}{\delta} \right). \quad (111)
$$

Since $\lambda^{i,-\hat{S}_i}$ and $\lambda^{i',-\hat{S}_{i'}}$ are losses on separate, unobserved tasks, they are dependent only through $U$. Therefore, they are conditionally independent given $(Z, \hat{S}, S^{\hat{S}})$. By the chain rule for the KL divergence, it follows that

$$
D(Q^{-\hat{S}} \,\|\, P^{-\hat{S}}) = \sum_{i=1}^{\hat{n}} D(Q^{i,-\hat{S}_i} \,\|\, P^{i,-\hat{S}_i}). \quad (112)
$$

By combining (107)-(112), we get, after some arithmetic, that with probability at least $1 - \delta$ under $(Z, \hat{S}, S)$,

$$
\left| L_{\mathcal{D}} - \widetilde{L} \right| \leq \sqrt{ \frac{\sum_{i=1}^{\hat{n}} 2D(Q^{i,-\hat{S}_i} \,\|\, P^{i,-\hat{S}_i}) + 2\log \left( \frac{\sqrt{n\hat{n}}}{\delta} \right)}{n\hat{n}-1} }. \quad (113)
$$

By the triangle inequality, $\left| L_{\mathcal{D}} - \widehat{L} \right| \leq \left| L_{\mathcal{D}} - \widetilde{L} \right| + \left| \widetilde{L} - \widehat{L} \right|$. By the union bound, (106) and (113) hold simultaneously with probability at least $1 - 2\delta$ under $(Z, \hat{S}, S)$. Therefore, with $\delta \to \delta/2$, they

hold simultaneously with probability at least $1 - \delta$. Thus, with probability at least $1 - \delta$ under the draw of $(Z, \hat{S}, S)$,

$$\left| L_{\mathcal{D}} - \widehat{L} \right| \leq \sqrt{\frac{\frac{1}{n}\sum_{j=1}^{n} 2D(Q_{j,S_j} \,||\, P_{j,S_j}) + 2\log\left(\frac{2n\sqrt{\hat{n}}}{\delta}\right)}{\hat{n} - 1}}$$
$$+ \sqrt{\frac{\sum_{i=1}^{\hat{n}} 2D(Q^{i,-\hat{S}_i} \,||\, P^{i,-\hat{S}_i}) + 2\log\left(\frac{2\sqrt{n\hat{n}}}{\delta}\right)}{n\hat{n} - 1}}. \quad (114)$$

$\square$

Having established Theorem 6, we now show how to use it to derive Theorem 4 under the assumption that the meta learner and base learner are invariant to the order of the samples.

*Proof of Theorem 4.* By the assumptions that the meta learner and base learner are invariant to the sample order and task index, we can reorder the data set so that $\max_j D(Q_{j,S_j} \,||\, P_{j,S_j}) = D(Q_{1,S_1} \,||\, P_{1,S_1})$. With this, $\frac{1}{n}\sum_{j=1}^{n} D(Q_{j,S_j} \,||\, P_{j,S_j}) \leq D(Q_{1,S_1} \,||\, P_{1,S_1})$. Similarly, we can reorder the data set so that $\max_i D(Q^{i,-\hat{S}_i} \,||\, P^{i,-\hat{S}_i}) = D(Q_\lambda^{1,-\hat{S}_1} \,||\, P_\lambda^{1,-\hat{S}_1})$, implying that $\sum_{i=1}^{\hat{n}} D(Q^{i,-\hat{S}_i} \,||\, P^{i,-\hat{S}_i}) \leq \hat{n} D(Q_\lambda^{1,-\hat{S}_1} \,||\, P_\lambda^{1,-\hat{S}_1})$. To obtain the final result, we note that for $n, \hat{n} \geq 2$, we have $1/(\hat{n}{-}1) \leq 2/\hat{n}$, $\log 2 \leq \log(n\sqrt{\hat{n}}/\delta)$, $1/(n{-}1) \leq 2/n$, $\log 2 \leq \log(\sqrt{n\hat{n}}/\delta)$, and $\log(\sqrt{n\hat{n}})/n\hat{n} \leq \log(\sqrt{n})/n$. Thus, we get the final result

$$\left| L_{\mathcal{D}} - \widehat{L} \right| \leq 2\sqrt{2}\sqrt{\frac{D(Q_{1,S_1}||P_{1,S_1}) + \log\left(\frac{n\sqrt{\hat{n}}}{\delta}\right)}{\hat{n}}} + 2\sqrt{2}\sqrt{\frac{D(Q^{1,-\hat{S}_1}||P^{1,-\hat{S}_1}) + \log\left(\frac{\sqrt{n}}{\delta}\right)}{n}}. \quad (115)$$

Thus, the bound holds with $C_1 = C_2 = 2\sqrt{2}$.

$\square$

While the simplifying assumption of invariance to the order of samples leads to a simpler result, it does not hold for all learning algorithms. Therefore, we will use the more general form given in (114) as the basis of Corollary 6.

Note that the first term of (115) diverges as $n \to \infty$. This counter-intuitive behavior, which requires both $n$ and $\hat{n}$ to be large for the bound to be nonvacuous, is common in PAC-Bayesian bounds for meta learning [29], and is seemingly an effect of the two-step approach. It is possible to obtain a different bound where this dependence is not explicit, similar to [28, 31], by simply not applying Jensen's inequality to the average over $j$ in (100) nor the average over $i$ in (107). However, the resulting environment-level KL divergence is different. In particular, if we were to use this alternative bound to derive minimax bounds in Section 4, the logarithmic dependence on $n$ would be embedded in this KL divergence. In the proof of Corollary 6, we point out where this difference would come into play.

We present the alternative bound in the following remark.

**Remark 1.** *Let $Q_S$ denote the conditional distribution of $\lambda_S$ given $(Z, \hat{S}, S)$ and let $P_S = \mathbb{E}_{\hat{S}}[Q_S]$. Then,*

$$\left| L_{\mathcal{D}} - \widehat{L} \right| \leq 2\sqrt{2}\sqrt{\frac{D(Q_S \,||\, P_S) + \log\left(\frac{\sqrt{\hat{n}}}{\delta}\right)}{\hat{n}}} + 2\sqrt{2}\sqrt{\frac{D(Q^{1,-\hat{S}_1} \,||\, P^{1,-\hat{S}_1}) + \log\left(\frac{\sqrt{n}}{\delta}\right)}{n}}. \quad (116)$$

*Proof.* The proof follows that of Theorem 4, so we only detail the differences: In (100), the average over $j$ is not moved outside the square when using Jensen's inequality; Donsker-Varadhan is now

used to perform a change of measure from $Q_S$ to $P_S$ in (101); and the union bound in (103) is no longer needed. $\square$

We now turn to Theorem 5.

*Proof of Theorem 5.* Recall that $Q$ denotes the conditional distribution of $\lambda$ given $(Z, \hat{S}, S)$, and that $P$ denotes $\mathbb{E}_{\hat{S},S}[Q]$. Let $\lambda$ be distributed according to $Q$. By Jensen's inequality,

$$\frac{n\hat{n}-1}{2}\Big(L_{\mathcal{D}}(Z,\hat{S},S)-\widehat{L}(Z,\hat{S},S)\Big)^2 \le \mathbb{E}_{\lambda}\left[\frac{n\hat{n}-1}{2}\left(\frac{1}{n\hat{n}}\sum_{i,j=1}^{\hat{n},n}\lambda_{j,-S_j^i}^{i,-\hat{S}_i} - \lambda_{j,S_j^i}^{i,\hat{S}_i}\right)^2\right]. \quad (117)$$

Next, let $\lambda'$ be distributed according to $P$. By Donsker-Varadhan's variational representation of the KL divergence,

$$\mathbb{E}_{\lambda}\left[\frac{n\hat{n}-1}{2}\left(\frac{1}{n\hat{n}}\sum_{i,j=1}^{\hat{n},n}\lambda_{j,-S_j^i}^{i,-\hat{S}_i} - \lambda_{j,S_j^i}^{i,\hat{S}_i}\right)^2\right]$$
$$\le D(Q\,||\,P) + \log\mathbb{E}_{\lambda'}\left[\frac{n\hat{n}-1}{2}\left(\frac{1}{n\hat{n}}\sum_{i,j=1}^{\hat{n},n}\lambda_{j,-S_j^i}'^{i,-\hat{S}_i} - \lambda_{j,S_j^i}'^{i,\hat{S}_i}\right)^2\right]. \quad (118)$$

By Markov's inequality, we conclude that with probability at least $1-\delta$ under the draw of $(Z, \hat{S}, S)$,

$$D(Q\,||\,P) + \log\mathbb{E}_{\lambda'}\left[\exp\left(\frac{n\hat{n}-1}{2}\left(\frac{1}{n\hat{n}}\sum_{i,j=1}^{\hat{n},n}\lambda_{j,-S_j^i}'^{i,-\hat{S}_i} - \lambda_{j,S_j^i}'^{i,\hat{S}_i}\right)^2\right)\right]$$
$$\le D(Q\,||\,P) + \log\mathbb{E}_{\lambda',Z,\hat{S},S}\left[\frac{1}{\delta}\exp\left(\frac{n\hat{n}-1}{2}\left(\frac{1}{n\hat{n}}\sum_{i,j=1}^{\hat{n},n}\lambda_{j,-S_j^i}'^{i,-\hat{S}_i} - \lambda_{j,S_j^i}'^{i,\hat{S}_i}\right)^2\right)\right]. \quad (119)$$

Now, note that $(\hat{S}, S)$ are independent from $\lambda'$, $Z$. Furthermore, $\lambda_{j,-S_j^i}'^{i,-\hat{S}_i} - \lambda_{j,S_j^i}'^{i,\hat{S}_i}$ is bounded to $[-1, 1]$, and $\mathbb{E}_{\hat{S}_i,S_j^i}\left[\lambda_{j,-S_j^i}'^{i,-\hat{S}_i} - \lambda_{j,S_j^i}'^{i,\hat{S}_i}\right] = 0$. Thus, $\frac{1}{n\hat{n}}\sum_{i,j=1}^{\hat{n},n}\lambda_{j,-S_j^i}'^{i,-\hat{S}_i} - \lambda_{j,S_j^i}'^{i,\hat{S}_i}$ is a $1/\sqrt{n\hat{n}}$-sub-Gaussian random variable, from which it follows that [38, Thm. 2.6.(IV)]

$$\log\mathbb{E}_{\lambda',Z,\hat{S},S}\left[\exp\left(\frac{n\hat{n}-1}{2}\left(\frac{1}{n\hat{n}}\sum_{i,j=1}^{\hat{n},n}\lambda_{j,-S_j^i}'^{i,-\hat{S}_i} - \lambda_{j,S_j^i}'^{i,\hat{S}_i}\right)^2\right)\right] \le \log\sqrt{n\hat{n}}. \quad (120)$$

By combining (117)-(120), we get

$$\frac{n\hat{n}-1}{2}\Big(L_{\mathcal{D}}(Z,\hat{S},S)-\widehat{L}(Z,\hat{S},S)\Big)^2 \le D(Q\,||\,P) + \log\frac{\sqrt{n\hat{n}}}{\delta}. \quad (121)$$

The desired result now follows after some arithmetic.

$\square$

## A.4 Proofs for Section 4

*Proof of Corollary 4.* We begin with (20). To establish this inequality, we bound the two sums on the right-hand side of (4) separately. First, by Jensen's inequality, we find that

$$\frac{1}{n\hat{n}} \sum_{i,j=1}^{\hat{n},n} \mathbb{E}_{Z,S_j^i}\left[\sqrt{I^{Z,S_j^i}\left(\lambda_{j,S_j^i}^i; \hat{S}_i\right)}\right] \leq \sqrt{\frac{1}{n\hat{n}} \sum_{i,j=1}^{\hat{n},n} I\left(\lambda_{j,S_j^i}^i; \hat{S}_i | Z, S\right)}. \tag{122}$$

Let $X_{j,l}^{i,k}$ denote the projection of $Z_{j,l}^{i,k}$ onto $\mathcal{X}$, i.e., $X_{j,l}^{i,k}$ contains the unlabelled instances from $Z$. The notation for $X_{j,l}^{i,k}$ is inherited from the notation for $Z_{j,l}^{i,k}$ introduced in Section 2. Let $f(\mathcal{A}(Z_{S^i}^{(i,k)}, R_i, \hat{\mathcal{A}}(Z_S^{\hat{S}}, \hat{R})), \cdot)$ denote the function from $\mathcal{F}$ that is selected by $\hat{\mathcal{A}}$ and $\mathcal{A}$ for task $(i,k)$ on the basis of $(Z, \hat{S}, S, R, \hat{R})$. We let $F_{j,S_j^i}^{i,k}$ denote the predicted label that the meta learner and the base learner produce for $X_{j,S_j^i}^{i,k}$. Furthermore, we let $F_{j,S_j^i}^i = (F_{j,S_j^i}^{i,0}, F_{j,S_j^i}^{i,1})$. Again, $F_{j,l}^{i,k}$ inherits the notational conventions that we use for $Z_{j,l}^{i,k}$. Note that, given $Z$, the losses $\lambda_{j,S_j^i}^i$ are a function of $F_{j,S_j^i}^i$. Thus, by the data-processing inequality,

$$\sqrt{\frac{1}{n\hat{n}} \sum_{i,j=1}^{\hat{n},n} I\left(\lambda_{j,S_j^i}^i; \hat{S}_i | Z, S\right)} \leq \sqrt{\frac{1}{n\hat{n}} \sum_{i,j=1}^{\hat{n},n} I\left(F_{j,S_j^i}^i; \hat{S}_i | Z, S\right)}. \tag{123}$$

Next, Let $h(\hat{\mathcal{A}}(Z_S^{\hat{S}}, \hat{R}), \cdot)$ denote the function from $\mathcal{H}$ that is selected by $\hat{\mathcal{A}}$ on the basis of $(Z, \hat{S}, S, R, \hat{R})$. We denote the representation that the meta learner induces on $X_{j,S_j^i}^i$ as $H_{j,S_j^i}^i$, the elements of which is given by, for $k \in \{0, 1\}$,

$$H_{j,S_j^i}^{i,k} = h(\hat{\mathcal{A}}(Z_S^{\hat{S}}, \hat{R}), X_{j,S_j^i}^{i,k}). \tag{124}$$

Note that, given $Z$, $S^i$ and $R_i$, the predictions in $F_{j,S_j^i}^i$ are a deterministic function of the intermediate representations $H_{j,S_j^i}^i$. Therefore, using the independence of $R_i$ and $\hat{S}_i$,

$$\sqrt{\frac{1}{n\hat{n}} \sum_{i,j=1}^{\hat{n},n} I\left(F_{j,S_j^i}^i; \hat{S}_i | Z, S\right)} \leq \sqrt{\frac{1}{n\hat{n}} \sum_{i,j=1}^{\hat{n},n} I\left(F_{j,S_j^i}^i; \hat{S}_i | Z, S, R_i\right)} \tag{125}$$

$$\leq \sqrt{\frac{1}{n\hat{n}} \sum_{i,j=1}^{\hat{n},n} I\left(H_{j,S_j^i}^i; \hat{S}_i | Z, S\right)}, \tag{126}$$

where $R_i$ disappears from the conditioning due to the independence of $H_{j,S_j^i}^i$ and $R_i$. Next, by adding random variables and using Lemma 3, we get

$$\sqrt{\frac{1}{n\hat{n}} \sum_{i,j=1}^{\hat{n},n} I\left(H_{j,S_j^i}^i; \hat{S}_i | Z, S\right)} \leq \sqrt{\frac{1}{n\hat{n}} \sum_{i,j=1}^{\hat{n},n} I\left(H_{j,S_j^i}; \hat{S}_i | Z, S\right)} \tag{127}$$

$$\leq \sqrt{\frac{\frac{1}{n} \sum_{j=1}^n I\left(H_{j,S_j}; \hat{S} | Z, S\right)}{\hat{n}}}. \tag{128}$$

For a given $j$, $Z$, and $S$, the $2\hat{n}$ inputs that give rise to $H_{j,S_j}$ are fixed. Thus, the number of possible different values that $H_{j,S_j}$ can take is at most $g_{\mathcal{H}}(2\hat{n})$, where $g_{\mathcal{H}}(\cdot)$ is the growth function of $\mathcal{H}$.

From this, it follows that

$$I\left(H_{j,S_j}; \hat{S}|Z, S\right) \leq \mathcal{H}\left(H_{j,S_j}|Z, S\right) \tag{129}$$

$$\leq \log g_{\mathcal{H}}(2\hat{n}) \tag{130}$$

$$\leq d_N \log\left(\binom{N}{2}\frac{2e\hat{n}}{d_N}\right). \tag{131}$$

Here, $\mathcal{H}(H_{j,S_j}|Z, S)$ denotes the conditional entropy of $H_{j,S_j}$ given $(Z, S)$, and the last inequality follows from Lemma 4. Since (129) does not depend on $j$, we find that

$$\sqrt{\frac{\frac{1}{n}\sum_{j=1}^n I\left(H_{j,S_j}; \hat{S}|Z, S\right)}{\hat{n}}} \leq \sqrt{\frac{d_N \log\left(\binom{N}{2}\frac{2e\hat{n}}{d_N}\right)}{\hat{n}}}. \tag{132}$$

By combining (122)-(132), we get

$$\frac{1}{n\hat{n}}\sum_{i,j=1}^{\hat{n},n} \mathbb{E}_{Z,S_j^i}\left[\sqrt{I^{Z,S_j^i}\left(\lambda_{j,S_j}^i; \hat{S}_i\right)}\right] \leq \sqrt{\frac{d_N \log\left(\binom{N}{2}\frac{2e\hat{n}}{d_N}\right)}{\hat{n}}}. \tag{133}$$

Next, we turn to the second sum on the right-hand side of (4). First, by Jensen's inequality,

$$\frac{1}{n\hat{n}}\sum_{i,j=1}^{\hat{n},n} \mathbb{E}_{Z,\hat{S}_i}\left[\sqrt{I^{Z,\hat{S}_i}\left(\lambda_j^{i,-\hat{S}_i}; S_j^{i,-\hat{S}_i}\right)}\right] \leq \sqrt{\frac{1}{n\hat{n}}\sum_{i,j=1}^{\hat{n},n} I\left(\lambda_j^{i,-\hat{S}_i}; S_j^{i,-\hat{S}_i}|Z, \hat{S}_i\right)}. \tag{134}$$

Note that, given $Z$, the losses $\lambda_j^{i,-\hat{S}_i}$ are a function of $F_j^{i,-\hat{S}_i}$. Therefore,

$$\sqrt{\frac{1}{n\hat{n}}\sum_{i,j=1}^{\hat{n},n} I\left(\lambda_j^{i,-\hat{S}_i}; S_j^{i,-\hat{S}_i}|Z, \hat{S}_i\right)} \leq \sqrt{\frac{1}{n\hat{n}}\sum_{i,j=1}^{\hat{n},n} I\left(F_j^{i,-\hat{S}_i}; S_j^{i,-\hat{S}_i}|Z, \hat{S}_i\right)} \tag{135}$$

$$\leq \sqrt{\frac{1}{n\hat{n}}\sum_{i,j=1}^{\hat{n},n} I\left(F^{i,-\hat{S}_i}; S_j^{i,-\hat{S}_i}|Z, \hat{S}_i\right)}, \tag{136}$$

where we used the fact that adding random variables cannot decrease mutual information. By the independence of $S_j^{i,-\hat{S}_i}$ for different $j$ and Lemma 3,

$$\sqrt{\frac{1}{n\hat{n}}\sum_{i,j=1}^{\hat{n},n} I\left(F^{i,-\hat{S}_i}; S_j^{i,-\hat{S}_i}|Z, \hat{S}_i\right)} \leq \sqrt{\frac{\frac{1}{\hat{n}}\sum_{i=1}^{\hat{n}} I\left(F^{i,-\hat{S}_i}; S^{i,-\hat{S}_i}|Z, \hat{S}_i\right)}{n}}. \tag{137}$$

Now, note that given $Z$ and $\hat{S}_i$, the $2n$ inputs that give rise to $F^{i,-\hat{S}_i}$ are fixed. Recall that $g_{\mathcal{F}}(\cdot)$ denotes the growth function of $\mathcal{F}$. Then,

$$\sqrt{\frac{\frac{1}{\hat{n}}\sum_{i=1}^{\hat{n}} I\left(F^{i,-\hat{S}_i}; S^{i,-\hat{S}_i}|Z, \hat{S}_i\right)}{n}} \leq \sqrt{\frac{\frac{1}{\hat{n}}\sum_{i=1}^{\hat{n}} \mathcal{H}\left(F^{i,-\hat{S}_i}|Z, \hat{S}_i\right)}{n}} \tag{138}$$

$$\leq \sqrt{\frac{\frac{1}{\hat{n}}\sum_{i=1}^{\hat{n}} g_{\mathcal{F}}(2n)}{n}} \tag{139}$$

$$\leq \sqrt{\frac{\frac{1}{\hat{n}}\sum_{i=1}^{\hat{n}} d_{\text{VC}} \log\left(\frac{2en}{d_{\text{VC}}}\right)}{n}}, \tag{140}$$

where we used Lemma 4. By combining (134)-(140), we get

$$\frac{1}{n\hat{n}} \sum_{i,j=1}^{\hat{n},n} \mathbb{E}_{Z,\hat{S}_i}\left[\sqrt{I^{Z,\hat{S}_i}\left(\lambda_j^{i,-\hat{S}_i}; S_j^{i,-\hat{S}_i}\right)}\right] \leq \sqrt{\frac{\frac{1}{\hat{n}}\sum_{i=1}^{\hat{n}} d_{\text{VC}} \log\left(\frac{2en}{d_{\text{VC}}}\right)}{n}}. \tag{141}$$

The result in (20) now follows by combining (4), (133) and (141).

We now turn to (21). First, by Jensen's inequality,

$$\frac{1}{n\hat{n}} \sum_{i,j=1}^{\hat{n},n} \mathbb{E}_Z\left[\sqrt{2I^Z(\lambda_j^i; \hat{S}_i, S_j^i)}\right] \leq \sqrt{\frac{1}{n\hat{n}} \sum_{i,j=1}^{\hat{n},n} 2I(\lambda_j^i; \hat{S}_i, S_j^i|Z)} \tag{142}$$

$$\leq \sqrt{\frac{1}{n\hat{n}} \sum_{i,j=1}^{\hat{n},n} 2I(\lambda; \hat{S}_i, S_j^i|Z)}, \tag{143}$$

where in the second step, we used that adding random variables does not decrease mutual information. Next, by the independence of the $S_j^i$ over $j$ and of the $(\hat{S}_i, S^i)$ over $i$,

$$\sqrt{\frac{1}{n\hat{n}} \sum_{i,j=1}^{\hat{n},n} 2I(\lambda; \hat{S}_i, S_j^i|Z)} \leq \sqrt{\frac{2I(\lambda; \hat{S}, S|Z)}{n\hat{n}}}. \tag{144}$$

Now, note that given $Z$, the losses $\lambda$ are a function of the predictions $F = \{F_j^i\}_{j=1:n}^{i=1:\hat{n}}$. Hence,

$$\sqrt{\frac{2I(\lambda; \hat{S}, S|Z)}{n\hat{n}}} \leq \sqrt{\frac{2I(F; \hat{S}, S|Z)}{n\hat{n}}} \tag{145}$$

$$\leq \sqrt{\frac{2I(F, H; \hat{S}, S|Z)}{n\hat{n}}} \tag{146}$$

where $H = \{H_j^i\}_{j=1:n}^{i=1:\hat{n}}$ and the second step follows by adding random variables. By the chain rule,

$$\sqrt{\frac{2I(F, H; \hat{S}, S|Z)}{n\hat{n}}} \leq \sqrt{\frac{2I(H; \hat{S}, S|Z) + 2I(F; \hat{S}, S|Z, H)}{n\hat{n}}} \tag{147}$$

$$\leq \sqrt{\frac{2\mathcal{H}(H|Z) + 2\mathcal{H}(F|Z, H)}{n\hat{n}}}. \tag{148}$$

Since $H$ is given by the elementwise application of some $h \in \mathcal{H}$ to $X = \{X_j^i\}_{j=1:n}^{1:\hat{n}}$, it can take at most $g_{\mathcal{H}}(4n\hat{n})$ different values, similar to previous arguments. This implies that

$$\mathcal{H}(H|Z) \leq \log(g_{\mathcal{H}}(4n\hat{n})) \leq d_N \log\left(\binom{N}{2} \frac{4en\hat{n}}{d_N}\right), \tag{149}$$

where the last inequality is again due to Lemma 4. Given $H$ and $Z$, the predictions $F$ can take at most $(g_{\mathcal{F}}(2n))^{2\hat{n}}$ different values, since the $2n$ inputs to each of the $2\hat{n}$ task-specific functions are fixed. This implies that

$$\mathcal{H}(F|H, Z) \leq 2\hat{n} \log(g_{\mathcal{F}}(2n)) \leq 2\hat{n}d_{\text{VC}} \log\left(\frac{2en}{d_{\text{VC}}}\right), \tag{150}$$

where we again used Lemma 4. The desired result follows by combining (6) with (142)-(150).

$\square$

*Proof of Corollary 5.* By the same steps as in (142)-(150), we find that

$$\frac{1}{n\hat{n}} \sum_{i,j=1}^{\hat{n},n} I(\lambda_j^i; \hat{S}_i, S_j^i | Z) \le d_N \log\left(\binom{N}{2} \frac{4en\hat{n}}{d_N}\right) + 2\hat{n}d_{\text{VC}} \log\left(\frac{2en}{d_{\text{VC}}}\right). \tag{151}$$

By combining this with (12), we find that

$$L_{\mathcal{D}} \le \frac{4d_N \log\left(\binom{N}{2} \frac{4en\hat{n}}{d_N}\right) + 8\hat{n}d_{\text{VC}} \log\left(\frac{2en}{d_{\text{VC}}}\right)}{n\hat{n}}. \tag{152}$$

This establishes the desired result. $\qquad\square$

*Proof of Corollary 6.* First, we establish (23). As mentioned in the proof of Theorem 4, we start the derivation from the more general bound given in (114) rather than the simplified bound given in (15). We begin by bounding $D(Q_{j,S_j} \| P_{j,S_j})$. Recall that $Q_{j,S_j}$ denotes the conditional distribution of $\lambda_{j,S_j}$ given $(Z, \hat{S}, S)$ and $P_{j,S_j} = \mathbb{E}_{\hat{S}}[Q_{j,S_j}]$. Let $\lambda_{j,S_j}$ be distributed according to $Q_{j,S_j}$. By Jensen's inequality,

$$D(Q_{j,S_j} \| P_{j,S_j}) = \mathbb{E}_{\lambda_{j,S_j}}\left[\log \frac{Q_{j,S_j}(\lambda_{j,S_j})}{P_{j,S_j}(\lambda_{j,S_j})}\right] \le \log \mathbb{E}_{\lambda_{j,S_j}}\left[\frac{Q_{j,S_j}(\lambda_{j,S_j})}{P_{j,S_j}(\lambda_{j,S_j})}\right]. \tag{153}$$

By Markov's inequality, with probability at least $1 - \delta$ under the draw of $(Z, \hat{S}, S)$,

$$D(Q_{j,S_j} \| P_{j,S_j}) = \mathbb{E}_{\lambda_{j,S_j}}\left[\log \frac{Q_{j,S_j}(\lambda_{j,S_j})}{P_{j,S_j}(\lambda_{j,S_j})}\right] \le \log\left(\frac{1}{\delta} \mathbb{E}_{\lambda_{j,S_j},Z,\hat{S},S}\left[\frac{Q_{j,S_j}(\lambda_{j,S_j})}{P_{j,S_j}(\lambda_{j,S_j})}\right]\right). \tag{154}$$

Since $\lambda_{j,S_j}$ is a discrete random variable, $Q_{j,S_j}(\lambda_{j,S_j}) \le 1$. Hence,

$$\log\left(\frac{1}{\delta} \mathbb{E}_{\lambda_{j,S_j},Z,\hat{S},S}\left[\frac{Q_{j,S_j}(\lambda_{j,S_j})}{P_{j,S_j}(\lambda_{j,S_j})}\right]\right) \le \log\left(\frac{1}{\delta} \mathbb{E}_{\lambda_{j,S_j},Z,\hat{S},S}\left[\frac{1}{P_{j,S_j}(\lambda_{j,S_j})}\right]\right). \tag{155}$$

Recall that $\mathbb{E}_{\hat{S}}[Q_{j,S_j}] = P_{j,S_j}$. Let $\lambda'_{j,S_j}$ be distributed according to $P_{j,S_j}$. Since the argument of the expectation is now independent of $\hat{S}$,

$$\log\left(\frac{1}{\delta} \mathbb{E}_{\lambda_{j,S_j},Z,\hat{S},S}\left[\frac{1}{P_{j,S_j}(\lambda_{j,S_j})}\right]\right) = \log\left(\frac{1}{\delta} \mathbb{E}_{\lambda'_{j,S_j},Z,S}\left[\frac{1}{P_{j,S_j}(\lambda_{j,S_j})}\right]\right) \tag{156}$$

$$\le \log\left(\frac{1}{\delta} \sup_{Z,S} \mathbb{E}_{\lambda'_{j,S_j}}\left[\frac{1}{P_{j,S_j}(\lambda_{j,S_j})}\right]\right). \tag{157}$$

Now, let $\Lambda_{j,S_j}(Z, S)$ denote the set of all possible values that $\lambda'_{j,S_j}$ can take given $(Z, S)$. Then,

$$\log\left(\frac{1}{\delta} \sup_{Z,S} \mathbb{E}_{\lambda'_{j,S_j}}\left[\frac{1}{P_{j,S_j}(\lambda_{j,S_j})}\right]\right) = \log\left(\frac{1}{\delta} \sup_{Z,S} \sum_{\lambda'_{j,S_j} \in \Lambda_{j,S_j}(Z,S)} \frac{P_{j,S_j}(\lambda_{j,S_j})}{P_{j,S_j}(\lambda_{j,S_j})}\right) \tag{158}$$

$$= \log\left(\frac{1}{\delta} \sup_{Z,S} |\Lambda_{j,S_j}(Z, S)|\right). \tag{159}$$

Now, note that since $\lambda_{j,S_j}$ is averaged over $R$, it is a function of $H_{j,S_j}$ given $(Z, S)$. Furthermore, the inputs $X_{j,S_j}$ are fixed. Therefore, as argued in the proof of Corollary 4, the number of different values that $H_{j,S_j}$ can take given $(Z, S)$ is at most $g_{\mathcal{H}}(2\hat{n})$.[3] Thus, by combining (153)-(159), we get

$$D(Q_{j,S_j} \| P_{j,S_j}) \le \log\left(\frac{g_{\mathcal{H}}(2\hat{n})}{\delta}\right) \le d_N \log\left(\binom{N}{2} \frac{2e\hat{n}}{d_N}\right) + \log \frac{1}{\delta}. \tag{160}$$

---

[3]If we had instead used the result of Remark 1 and followed analogous steps, we would instead get $g_{\mathcal{H}}(2n\hat{n})$ as an upper bound of the KL divergence.

Next, we turn to $D(Q^{-\hat{S}} \| P^{-\hat{S}})$. Recall that $Q^{-\hat{S}}$ denotes the conditional distribution of $\lambda^{-\hat{S}}$ given $(Z, \hat{S}, S)$, and $P^{-\hat{S}} = \mathbb{E}_{S^{-\hat{s}}} \left[ Q^{-\hat{S}} \right]$. Let $\lambda^{-\hat{S}}$ be distributed according to $Q^{-\hat{S}}$. Again, by Jensen's inequality,

$$D(Q^{-\hat{S}} \| P^{-\hat{S}}) = \mathbb{E}_{\lambda^{-\hat{s}}} \left[ \log \frac{Q^{-\hat{S}}(\lambda^{-\hat{S}})}{P^{-\hat{S}}(\lambda^{-\hat{S}})} \right] \leq \log \mathbb{E}_{\lambda^{-\hat{s}}} \left[ \frac{Q^{-\hat{S}}(\lambda^{-\hat{S}})}{P^{-\hat{S}}(\lambda^{-\hat{S}})} \right]. \tag{161}$$

By Markov's inequality, with probability at least $1 - \delta$ under the draw of $(Z, \hat{S}, S)$,

$$D(Q^{-\hat{S}} \| P^{-\hat{S}}) = \mathbb{E}_{\lambda^{-\hat{s}}} \left[ \log \frac{Q^{-\hat{S}}(\lambda^{-\hat{S}})}{P^{-\hat{S}}(\lambda^{-\hat{S}})} \right] \leq \log \left( \frac{1}{\delta} \mathbb{E}_{\lambda^{-\hat{s}}, Z, \hat{S}, S} \left[ \frac{Q^{-\hat{S}}(\lambda^{-\hat{S}})}{P^{-\hat{S}}(\lambda^{-\hat{S}})} \right] \right). \tag{162}$$

Since $\lambda^{-\hat{S}}$ is a discrete random variable, $Q^{-\hat{S}}(\lambda^{-\hat{S}}) \leq 1$. Therefore,

$$\log \left( \frac{1}{\delta} \mathbb{E}_{\lambda^{-\hat{s}}, Z, \hat{S}, S} \left[ \frac{Q^{-\hat{S}}(\lambda^{-\hat{S}})}{P^{-\hat{S}}(\lambda^{-\hat{S}})} \right] \right) \leq \log \left( \frac{1}{\delta} \mathbb{E}_{\lambda^{-\hat{s}}, Z, \hat{S}, S} \left[ \frac{1}{P^{-\hat{S}}(\lambda^{-\hat{S}})} \right] \right). \tag{163}$$

Now, let $\lambda'^{-\hat{S}}$ be distributed according to $P^{-\hat{S}}$. Since the argument of the expectation is now independent of $S^{-\hat{S}}$,

$$\log \left( \frac{1}{\delta} \mathbb{E}_{\lambda^{-\hat{s}}, Z, \hat{S}, S} \left[ \frac{1}{P^{-\hat{S}}(\lambda^{-\hat{S}})} \right] \right) = \log \left( \frac{1}{\delta} \mathbb{E}_{\lambda'^{-\hat{s}}, Z, \hat{S}, S^{\hat{S}}} \left[ \frac{1}{P^{-\hat{S}}(\lambda^{-\hat{S}})} \right] \right) \tag{164}$$

$$\leq \log \left( \frac{1}{\delta} \sup_{Z, \hat{S}, S^{\hat{s}}} \mathbb{E}_{\lambda'^{-\hat{s}}} \left[ \frac{1}{P^{-\hat{S}}(\lambda^{-\hat{S}})} \right] \right). \tag{165}$$

Now, let $\Lambda^{-\hat{S}}(Z, S)$ denote the set of all possible values that $\lambda'^{-\hat{S}}$ can take given $(Z, \hat{S}, S^{\hat{S}})$. Then,

$$\log \left( \frac{1}{\delta} \sup_{Z, \hat{S}, S^{\hat{s}}} \mathbb{E}_{\lambda'^{-\hat{s}}} \left[ \frac{1}{P^{-\hat{S}}(\lambda^{-\hat{S}})} \right] \right) = \log \left( \frac{1}{\delta} \sup_{Z, \hat{S}, S^{\hat{s}}} \sum_{\lambda'^{-\hat{s}} \in \Lambda^{-\hat{s}}(Z,S)} \frac{P^{-\hat{S}}(\lambda^{-\hat{S}})}{P^{-\hat{S}}(\lambda^{-\hat{S}})} \right) \tag{166}$$

$$= \log \left( \frac{1}{\delta} \sup_{Z, \hat{S}, S^{\hat{s}}} \left| \Lambda^{-\hat{S}}(Z, S) \right| \right). \tag{167}$$

Note that, given $Z$, the losses $\lambda^{-i, \hat{S}_i}$ are a function of the predictions $F^{i, -\hat{S}_i}$. Furthermore, given $(Z, \hat{S}, S^{\hat{S}})$, the inputs $H^{i, -\hat{S}_i}$ are fixed. This is the case since $U$ is independent from $S^{-\hat{S}}$. Thus, similar to previous arguments, given $(Z, \hat{S}, S^{\hat{S}})$, $F^{i, -\hat{S}_i}$ can take at most $g_{\mathcal{F}}(2n)$ different values for each $i$. Therefore, $F^{-\hat{S}}$ can take at most $g_{\mathcal{F}}(2n)^{\hat{n}}$ values. Thus, by combining (161)-(167), we get

$$D(Q^{-\hat{S}} \| P^{-\hat{S}}) \leq \log \left( \frac{g_{\mathcal{F}}(2n)^{\hat{n}}}{\delta} \right) \leq \hat{n} d_{\text{VC}} \log \left( \frac{2en}{d_{\text{VC}}} \right) + \log \frac{1}{\delta}, \tag{168}$$

where we used Lemma 4. Thus, by using a union bound, we can combine (114), (160), and (168), with $\delta \to \delta/3$, to conclude that with probability at least $1 - \delta$ under the draw of $(Z, \hat{S}, S)$,

$$\left| L_{\mathcal{D}} - \hat{L} \right| \leq \sqrt{\frac{2 d_N \log \left( \binom{N}{2} \frac{2e\hat{n}}{d_N} \right) + \log \frac{3}{\delta} + 2 \log \left( \frac{6n\sqrt{\hat{n}}}{\delta} \right)}{\hat{n} - 1}}$$

$$+ \sqrt{\frac{2\hat{n} d_{\text{VC}} \log \left( \frac{2en}{d_{\text{VC}}} \right) + \log \frac{3}{\delta} + 2 \log \left( \frac{6\sqrt{n\hat{n}}}{\delta} \right)}{n\hat{n} - 1}}. \tag{169}$$

Under the assumption that $n, \hat{n} \geq 2$, by similar arguments as in the proof of Theorem 4, we find that, for some constants $C_1$ and $C_2$,

$$\left|L_{\mathcal{D}} - \widehat{L}\right| \leq C_1 \sqrt{\frac{d_N \log\left(\binom{N}{2} \frac{\hat{n}}{d_N}\right) + \log\left(\frac{n\sqrt{\hat{n}}}{\delta}\right)}{\hat{n}}} + C_2 \sqrt{\frac{d_{\text{VC}} \log\left(\frac{n}{d_{\text{VC}}}\right) + \log\left(\frac{\sqrt{n}}{\delta}\right)}{n}}. \quad (170)$$

This establishes (23).

We now turn to (24). Let $\lambda$ be distributed according to $Q$. First, by Jensen's inequality,

$$D(Q \,\|\, P) = \mathbb{E}_\lambda\left[\log \frac{Q(\lambda)}{P(\lambda)}\right] \leq \log \mathbb{E}_\lambda\left[\frac{Q(\lambda)}{P(\lambda)}\right]. \quad (171)$$

By Markov's inequality, with probability at least $1 - \delta$ under the draw of $(Z, \hat{S}, S)$,

$$D(Q \,\|\, P) = \mathbb{E}_\lambda\left[\log \frac{Q(\lambda)}{P(\lambda)}\right] \leq \log\left(\frac{1}{\delta} \mathbb{E}_{\lambda, Z, \hat{S}, S}\left[\frac{Q(\lambda)}{P(\lambda)}\right]\right). \quad (172)$$

Since $\lambda$ is a discrete random variable, $Q(\lambda) \leq 1$. Hence,

$$D(Q \,\|\, P) = \mathbb{E}_\lambda\left[\log \frac{Q(\lambda)}{P(\lambda)}\right] \leq \log\left(\frac{1}{\delta} \mathbb{E}_{\lambda, Z, \hat{S}, S}\left[\frac{1}{P(\lambda)}\right]\right). \quad (173)$$

Recall that $\mathbb{E}_{\hat{S}, S}[Q] = P$. Let $\lambda'$ be distributed according to $P$. Since the argument of the expectation is now independent of $(\hat{S}, S)$,

$$\log\left(\frac{1}{\delta} \mathbb{E}_{\lambda, Z, \hat{S}, S}\left[\frac{1}{P(\lambda)}\right]\right) \leq \log\left(\frac{1}{\delta} \mathbb{E}_{\lambda', Z}\left[\frac{1}{P(\lambda')}\right]\right) \quad (174)$$

$$\leq \log\left(\frac{1}{\delta} \sup_Z \mathbb{E}_{\lambda'}\left[\frac{1}{P(\lambda')}\right]\right). \quad (175)$$

Let $\Lambda(Z)$ denote the set of all possible values that $\lambda'$ can take given $Z$. Then,

$$\log\left(\frac{1}{\delta} \sup_Z \mathbb{E}_{\lambda'}\left[\frac{1}{P(\lambda')}\right]\right) = \log\left(\frac{1}{\delta} \sup_Z \sum_{\lambda' \in \Lambda(Z)} \frac{P(\lambda')}{P(\lambda')}\right) \quad (176)$$

$$= \log\left(\frac{1}{\delta} \sup_Z |\Lambda(Z)|\right). \quad (177)$$

Since the map from predictions to losses is surjective, $|\Lambda(Z)|$ is bounded by the number of possible predictions $F$ given $Z$. We can bound this as follows. First, the number of possible different values for $H$ given $Z$ is at most $g_{\mathcal{H}}(4n\hat{n})$. Given a fixed $H$, the number of possible values that $F$ can take is at most $(g_{\mathcal{F}}(2n))^{2\hat{n}}$, since the $2n$ inputs to each of the $2\hat{n}$ task-specific functions are fixed. Therefore, the total number of possible values for $F$ given $Z$ is at most $g_{\mathcal{H}}(4n\hat{n})(g_{\mathcal{F}}(2n))^{2\hat{n}}$. Hence,

$$\log\left(\frac{1}{\delta} \sup_Z |\Lambda(Z)|\right) \leq \log(g_{\mathcal{H}}(4n\hat{n})) + 2\hat{n} \log(g_{\mathcal{F}}(2n)) + \log \frac{1}{\delta} \quad (178)$$

$$\leq d_N \log\left(\binom{N}{2} \frac{4en\hat{n}}{d_N}\right) + 2\hat{n} d_{\text{VC}} \log\left(\frac{2en}{d_{\text{VC}}}\right) + \log \frac{1}{\delta} \quad (179)$$

where we used Lemma 4. Substituting this into (18), using a union bound and letting $\delta \to \delta/2$, we find that with probability at least $1 - \delta$ under the draw of $(Z, \hat{S}, S)$,

$$
\left| L_{\mathcal{D}}(Z, \hat{S}, S) - \widehat{L}(Z, \hat{S}, S) \right|
$$

$$
\leq \sqrt{\frac{2d_N \log\left(\binom{N}{2}\frac{4en\hat{n}}{d_N}\right) + 4\hat{n}d_{\mathrm{VC}} \log\left(\frac{2en}{d_{\mathrm{VC}}}\right) + 2\log\frac{2}{\delta} + 2\log\left(\frac{2\sqrt{n\hat{n}}}{\delta}\right)}{n\hat{n} - 1}}. \quad (180)
$$

Assuming that $n, \hat{n} \geq 2$, the desired result in (21) follows by upper-bounding constants by using similar arguments as in the proof of Theorem 4.

$\square$

# B  Bound for the Excess Risk

We now turn to excess risk bounds. In Corollary 7, we present the formal statement of the excess risk bound in Section 4.2. In Corollary 8, we state an excess risk bound for a randomly drawn new task, which obviates the need of a task diversity assumption.

In order to derive excess risk bounds, we need to introduce some technical tools. First, we need to consider *oracle* algorithms, that is, algorithms that output minimizers of the population loss. Specifically, the oracle meta learner knows the task distribution $\mathcal{D}$, while the oracle base learner knows the indexed set of in-task distributions $\{D_\tau : \tau \in \mathcal{T}\}$. While these algorithms have access to the data distributions, and are thus not of practical interest, they are useful as a proof technique, and can be analyzed in the same way as realistic algorithms. Second, in order to allow the oracle base learner to minimize the population loss for a given task, we need to extend the input to the base learner to include the identity of the task $\tau \in \mathcal{T}$. Thus, the base learner is a mapping $\mathcal{A} : \mathcal{Z}^n \times \mathcal{T} \times \mathcal{R} \times \mathcal{U} \to \mathcal{W}$. For the case of a base learner that minimizes the empirical risk, the task identity is irrelevant, so the input from $\mathcal{T}$ does not affect the output. Conversely, for an oracle base learner, the training samples are irrelevant, so only the input from $\mathcal{T}$ affects the output. Finally, our information-theoretic bounds pertain to a test loss, rather than the population loss. While these are equal for average bounds, there is a small discrepancy for the high-probability bounds. In order to handle excess risk bounds and oracle algorithms that depend on the population loss, we need to convert between the two by using a Hoeffding bound, as discussed in [23, Thm. 3]. The extra terms that this additional step leads to are typically negligible compared to the dominant complexity terms.

For concreteness, we focus only on high-probability excess risk bounds derived on the basis of the one-step square-root bound in Corollary 6. However, note that excess risk bounds based on the other high-probability bounds can be obtained by suitably substituting these alternative bounds in the proofs. Average excess risk bounds can also be derived by an analogous procedure. First, using the task diversity assumption of [5], we derive an excess risk bound for a fixed target task.

**Corollary 7.** *Consider the setting of Corollary 6 and a fixed task $\tau_0$. Let $Z^0 \in \mathcal{Z}^{2 \times m}$ be a matrix of $2m$ samples generated independently according to $\mathcal{D}_{\tau_0}$, the data distribution for task $\tau_0$. Let $S^0$ be an $m$-dimensional random vector with elements generated independently from a $\mathrm{Bern}(1/2)$ distribution, and let the training set $Z^0_{S^0}$ and test set $Z^0_{-S^0}$ be constructed in the same way as the training and test sets for tasks $1, \ldots, \hat{n}$. To simplify notation, let $Z^{0,0} = Z^{0,1} = Z^0$. Denote the population loss for the $i$th observed task when using the base learner $\mathcal{A}'$ with the representation $h'$ as*

$$
L_{\mathcal{D}}(i, \mathcal{A}', h') = \mathbb{E}_{R_i, \tilde{Z} \sim \mathcal{D}_{\tau_{i, \hat{S}_i}}} \left[ \ell\left(\mathcal{A}'\left(Z_{S^i}^{i, \hat{S}_i}, \tau_{i, \hat{S}_i}, R_i, h'\right), \tilde{Z}\right) \right]. \quad (181)
$$

*Similarly, denote the population loss for the $i$th unobserved task when using the base learner $\mathcal{A}'$ with the representation $h'$ as*

$$
L_{\mathcal{D}}(-i, \mathcal{A}', h') = \mathbb{E}_{R_i, \tilde{Z} \sim \mathcal{D}_{\tau_{i, -\hat{S}_i}}} \left[ \ell\left(\mathcal{A}'\left(Z_{S^i}^{i, -\hat{S}_i}, \tau_{i, -\hat{S}_i}, R_i, h'\right), \tilde{Z}\right) \right]. \quad (182)
$$

Let $\mathcal{A}^*$ denote an oracle learner that satisfies $L_\mathcal{D}(i, \mathcal{A}^*, h') = \min_{\mathcal{A}'} L_\mathcal{D}(i, \mathcal{A}', h')$ for all $h'$. Assume that $h^* = \arg\min_{h'} \min_{\mathcal{A}'} L_\mathcal{D}(i, \mathcal{A}', h')$ for all $\tau_{i,\hat{S}_i}$ and that $h^* \in \mathcal{H}$. Thus, the same representation $h^*$ is optimal for all tasks. Let $\hat{\mathcal{A}}$ and $\mathcal{A}$ be empirical risk minimizers, and let $\hat{h} = \hat{\mathcal{A}}(\hat{R}, Z_S^{\hat{S}})$. Finally, assume that the supersample satisfies a task-diversity assumption, so that for some $\nu$ and $\epsilon$,

$$\sup_{\tau_0} L_\mathcal{D}(0, \mathcal{A}^*, \hat{h}) - L_\mathcal{D}(0, \mathcal{A}^*, h^*) \leq \nu^{-1}\left( L_\mathcal{D}(-1{:}\hat{n}, \mathcal{A}^*, \hat{h}) - L_\mathcal{D}(-1{:}\hat{n}, \mathcal{A}^*, h^*) \right) + \epsilon. \quad (183)$$

Then, there exist constants $C_1$ and $C_2$ such that, with probability at least $1 - \delta$ under the draw of $(Z, \hat{S}, S, Z^0, S^0)$, we have

$$L_\mathcal{D}(0, \mathcal{A}, \hat{h}) - L_\mathcal{D}(0, \mathcal{A}^*, h^*) \leq C_1 \sqrt{\frac{d_{\text{VC}} \log\left(\frac{\sqrt{m}}{d_{\text{vc}}}\right) + \log\left(\frac{\sqrt{m}}{\delta}\right)}{m}}$$

$$+ C_2 \nu^{-1} \sqrt{\frac{d_N \log\left(\binom{N}{2}\frac{n\hat{n}}{d_N}\right) + \hat{n} d_{\text{VC}} \log\left(\frac{n}{d_{\text{vc}}}\right) + \log\left(\frac{\sqrt{n\hat{n}}}{\delta}\right)}{n\hat{n}}} + \epsilon. \quad (184)$$

*Proof.* We will use the following shorthands. When using the algorithm $\mathcal{A}'$ for task $i$ based on the representation $h'$, we let $L_\mathcal{D}(i, \mathcal{A}', h')$ denote the population loss, $\hat{L}(i, \mathcal{A}', h')$ denote the training loss, and $\widetilde{L}(i, \mathcal{A}', h')$ denote the test loss on a test set of the same size as the training set. Formally,

$$L_\mathcal{D}(i, \mathcal{A}', h') = \mathbb{E}_{R_i, \tilde{Z} \sim \mathcal{D}_{\tau_{i,\hat{S}_i}}}\left[ \ell\left( \mathcal{A}'\left( Z_{S^i}^{i,\hat{S}_i}, \tau_{i,\hat{S}_i}, R_i, h' \right), \tilde{Z} \right) \right], \quad (185)$$

$$\hat{L}(i, \mathcal{A}', h') = \frac{1}{n} \sum_{j=1}^{n} \mathbb{E}_{R_i}\left[ \ell\left( \mathcal{A}'\left( Z_{S^i}^{i,\hat{S}_i}, \tau_{i,\hat{S}_i}, R_i, h' \right), Z_{j,S_j^i}^{i,\hat{S}_i} \right) \right], \quad (186)$$

$$\widetilde{L}(i, \mathcal{A}', h') = \frac{1}{n} \sum_{j=1}^{n} \mathbb{E}_{R_i}\left[ \ell\left( \mathcal{A}'\left( Z_{S^i}^{i,\hat{S}_i}, \tau_{i,\hat{S}_i}, R_i, h' \right), Z_{j,-S_j^i}^{i,\hat{S}_i} \right) \right]. \quad (187)$$

As a shorthand, we let $L_\mathcal{D}(1{:}\hat{n}, \mathcal{A}', h') = \frac{1}{\hat{n}}\sum_{i=1}^{\hat{n}} L_\mathcal{D}(i, \mathcal{A}', h')$, and we use the same convention for $\hat{L}(1{:}\hat{n}, \mathcal{A}', h')$ and $\widetilde{L}(1{:}\hat{n}, \mathcal{A}', h')$. Furthermore, to indicate losses on unobserved tasks we negate the task index. Thus,

$$\hat{L}(-i, \mathcal{A}', h') = \frac{1}{n} \sum_{j=1}^{n} \mathbb{E}_{R_i}\left[ \ell\left( \mathcal{A}'\left( Z_{S^i}^{i,-\hat{S}_i}, \tau_{i,-\hat{S}_i}, R_i, h' \right), Z_{j,S_j^i}^{i,-\hat{S}_i} \right) \right], \quad (188)$$

with analogous notation for the test and population losses.

The base learner $\mathcal{A}$ that we consider is an empirical risk minimizer, which satisfies for all $h'$

$$\hat{L}(i, \mathcal{A}, h') = \min_{\mathcal{A}'} \hat{L}(i, \mathcal{A}', h'). \quad (189)$$

For our analysis, we use an oracle learner $\mathcal{A}^*$, which outputs the minimizer of the population loss for the given task. While this is not a realistic learning algorithm in practice, as it depends on the data distribution, it is useful as an analysis tool. Formally, for all $h'$,

$$L_\mathcal{D}(i, \mathcal{A}^*, h') = \min_{\mathcal{A}'} L_\mathcal{D}(i, \mathcal{A}', h'). \quad (190)$$

Finally, we let $\hat{h}$ be a representation that minimizes the empirical risk over the $\hat{n}$ training tasks and $h^*$ be an optimal representation, i.e.

$$\hat{h} \in \arg\min_{h'} \hat{L}(1{:}\hat{n}, \mathcal{A}, h'). \quad (191)$$

$$h^* \in \arg\min_{h'} L_\mathcal{D}(i, \mathcal{A}^*, h'). \quad (192)$$

By assumption, $h^*$ is the same for any task $\tau_{i,k}$.

In the proof, we need to convert between test losses and population losses. By definition, test data is independent from the hypothesis, so standard concentration inequalities can be applied to bound the difference between the test and population loss. The following lemma follows immediately from Hoeffding's inequality [38, Prop. 2.5], as argued in [23, Thm. 3].

**Lemma 5.** *Let $\widetilde{L}(i, \mathcal{A}', h')$ be a test loss based on $m$ samples. Then, with probability at least $1 - \delta$,*

$$\left| \widetilde{L}(i, \mathcal{A}', h') - L_{\mathcal{D}}(i, \mathcal{A}', h') \right| \le \sqrt{\frac{\log \frac{2}{\delta}}{2m}}. \tag{193}$$

*Proof.* The test loss $\widetilde{L}(i, \mathcal{A}', h')$ is the average of $m$ independent samples of a bounded random variable with mean $L_{\mathcal{D}}(i, \mathcal{A}', h')$. Therefore, the result follows by Hoeffding's inequality [38, Prop. 2.5]. $\qquad\square$

This result allows us to convert between test losses and population losses at the cost of a term that is typically negligible in comparison to the complexity terms.

With these tools and notations in place, we are ready to derive excess risk bounds. The aim is to upper-bound the excess risk by an expression consisting of differences between training and test losses, for which we can apply our generalization bounds. Starting from the excess risk on task $\tau_0$, which is our fixed target task, we get

$$L_{\mathcal{D}}(0, \mathcal{A}, \hat{h}) - L_{\mathcal{D}}(0, \mathcal{A}^*, h^*) = L_{\mathcal{D}}(0, \mathcal{A}, \hat{h}) - L_{\mathcal{D}}(0, \mathcal{A}^*, \hat{h}) + L_{\mathcal{D}}(0, \mathcal{A}^*, \hat{h}) - L_{\mathcal{D}}(0, \mathcal{A}^*, h^*)$$

$$\le L_{\mathcal{D}}(0, \mathcal{A}, \hat{h}) - L_{\mathcal{D}}(0, \mathcal{A}^*, \hat{h}) + D. \tag{194}$$

Here, $D = \sup_{\tau_0} L_{\mathcal{D}}(0, \mathcal{A}^*, \hat{h}) - L_{\mathcal{D}}(0, \mathcal{A}^*, h^*)$ is the worst-case representation difference [5], which we will later bound using a task diversity assumption. Next, by Lemma 5, with probability at least $1 - 2\delta$,

$$L_{\mathcal{D}}(0, \mathcal{A}, \hat{h}) - L_{\mathcal{D}}(0, \mathcal{A}^*, \hat{h}) + D \le \widetilde{L}(0, \mathcal{A}, \hat{h}) - \widetilde{L}(0, \mathcal{A}^*, \hat{h}) + D + 2\sqrt{\frac{\log \frac{2}{\delta}}{m}}. \tag{195}$$

Next, we use the risk decomposition

$$\widetilde{L}(0, \mathcal{A}, \hat{h}) - \widetilde{L}(0, \mathcal{A}^*, \hat{h}) \tag{196}$$

$$= \widetilde{L}(0, \mathcal{A}, \hat{h}) - \hat{L}(0, \mathcal{A}, \hat{h}) + \hat{L}(0, \mathcal{A}, \hat{h}) - \hat{L}(0, \mathcal{A}^*, \hat{h}) + \hat{L}(0, \mathcal{A}^*, \hat{h}) - \widetilde{L}(0, \mathcal{A}^*, \hat{h}) \tag{197}$$

$$\le \widetilde{L}(0, \mathcal{A}, \hat{h}) - \hat{L}(0, \mathcal{A}, \hat{h}) + \hat{L}(0, \mathcal{A}^*, \hat{h}) - \widetilde{L}(0, \mathcal{A}^*, \hat{h}), \tag{198}$$

where the last step follows because $\hat{L}(0, \mathcal{A}, \hat{h}) \le \hat{L}(0, \mathcal{A}^*, \hat{h})$, since $\mathcal{A}$ is an empirical risk minimizer. Notice that the resulting expression is the difference between test and training losses on task $\tau_0$ for two different algorithms. These terms are simply the generalization gaps for a conventional learning setting. These terms can be bounded by applying Corollary 6, but for the case where $\hat{n} = 1$ and $\mathcal{H} = \{\hat{h}\}$, which implies that $d_N = 0$. We conclude that there exists a constant $C_1$ such that, with probability at least $1 - \delta$,

$$\widetilde{L}(0, \mathcal{A}, \hat{h}) - \hat{L}(0, \mathcal{A}, \hat{h}) + \hat{L}(0, \mathcal{A}^*, \hat{h}) - \widetilde{L}(0, \mathcal{A}^*, \hat{h}) \le C_1 \sqrt{\frac{d_{\mathrm{VC}} \log\left(\frac{\sqrt{m}}{d_{\mathrm{VC}}}\right) + \log\left(\frac{\sqrt{m}}{\delta}\right)}{m}}. \tag{199}$$

It remains to bound $D$. First, by the task diversity assumption,

$$D = \sup_{\tau_0} L_{\mathcal{D}}(0, \mathcal{A}^*, \hat{h}) - L_{\mathcal{D}}(0, \mathcal{A}^*, h^*) \tag{200}$$

$$\le \nu^{-1}\left( L_{\mathcal{D}}(-1{:}\hat{n}, \mathcal{A}^*, \hat{h}) - L_{\mathcal{D}}(-1{:}\hat{n}, \mathcal{A}^*, h^*) \right) + \epsilon. \tag{201}$$

We note here that, while the way that [5] uses the assumption of task diversity requires that the difference between the minimum population losses for task $\tau_0$ based on $\hat{h}$ and $h^*$ is controlled by the corresponding risks for tasks $1{:}\hat{n}$, i.e., the tasks upon which $\hat{h}$ is chosen, we instead assume that it is controlled by the corresponding losses for tasks $-1{:}\hat{n}$, i.e., tasks that are independent from $\hat{h}$. In this sense, the diversity assumption that we use is arguably weaker.

By a risk decomposition, we get

$$L_{\mathcal{D}}(-1{:}\hat{n}, \mathcal{A}^*, \hat{h}) - L_{\mathcal{D}}(-1{:}\hat{n}, \mathcal{A}^*, h^*) \tag{202}$$

$$=L_{\mathcal{D}}(-1{:}\hat{n}, \mathcal{A}^*, \hat{h}) - L_{\mathcal{D}}(-1{:}\hat{n}, \mathcal{A}, \hat{h}) + L_{\mathcal{D}}(-1{:}\hat{n}, \mathcal{A}, \hat{h}) - L_{\mathcal{D}}(-1{:}\hat{n}, \mathcal{A}^*, h^*) \tag{203}$$

$$\leq L_{\mathcal{D}}(-1{:}\hat{n}, \mathcal{A}, \hat{h}) - L_{\mathcal{D}}(-1{:}\hat{n}, \mathcal{A}^*, h^*), \tag{204}$$

where the last step follows since $L_{\mathcal{D}}(-1{:}\hat{n}, \mathcal{A}^*, \hat{h}) \leq L_{\mathcal{D}}(-1{:}\hat{n}, \mathcal{A}, \hat{h})$. By Lemma 5, with probability $1 - 2\delta$,

$$L_{\mathcal{D}}(-1{:}\hat{n}, \mathcal{A}, \hat{h}) - L_{\mathcal{D}}(-1{:}\hat{n}, \mathcal{A}^*, h^*) \leq \widetilde{L}(-1{:}\hat{n}, \mathcal{A}, \hat{h}) - \widetilde{L}(-1{:}\hat{n}, \mathcal{A}^*, h^*) + 2\sqrt{\frac{\log\frac{2}{\delta}}{2n}}. \tag{205}$$

By a risk decomposition, we find that

$$\widetilde{L}(-1{:}\hat{n}, \mathcal{A}, \hat{h}) - \widetilde{L}(-1{:}\hat{n}, \mathcal{A}^*, h^*)$$

$$\leq \widetilde{L}(-1{:}\hat{n}, \mathcal{A}, \hat{h}) - \hat{L}(1{:}\hat{n}, \mathcal{A}, \hat{h}) + \hat{L}(1{:}\hat{n}, \mathcal{A}, \hat{h}) - \hat{L}(1{:}\hat{n}, \mathcal{A}^*, h^*) + \hat{L}(1{:}\hat{n}, \mathcal{A}^*, h^*) - \widetilde{L}(-1{:}\hat{n}, \mathcal{A}^*, h^*)$$

$$\leq \widetilde{L}(-1{:}\hat{n}, \mathcal{A}, \hat{h}) - \hat{L}(1{:}\hat{n}, \mathcal{A}, \hat{h}) + \hat{L}(1{:}\hat{n}, \mathcal{A}^*, h^*) - \widetilde{L}(-1{:}\hat{n}, \mathcal{A}^*, h^*), \tag{206}$$

where the last step follows since $\hat{L}(1{:}\hat{n}, \mathcal{A}, \hat{h}) \leq \hat{L}(1{:}\hat{n}, \mathcal{A}^*, h^*)$. Now, notice that the resulting expression consists of the differences between the unobserved test losses and observed training losses for two different learning algorithms. This means that we can apply Corollary 6 to find that there exists a constant $C_2$ such that, with probability at least $1 - \delta$,

$$\widetilde{L}(-1{:}\hat{n}, \mathcal{A}, \hat{h}) - \hat{L}(1{:}\hat{n}, \mathcal{A}, \hat{h}) + \hat{L}(1{:}\hat{n}, \mathcal{A}^*, h^*) - \widetilde{L}(-1{:}\hat{n}, \mathcal{A}^*, h^*)$$

$$\leq C_2 \sqrt{\frac{d_N \log\left(\binom{N}{2}\frac{n\hat{n}}{d_N}\right) + \hat{n}d_{\mathrm{VC}} \log\left(\frac{n}{d_{\mathrm{VC}}}\right) + \log\left(\frac{\sqrt{n\hat{n}}}{\delta}\right)}{n\hat{n}}}. \tag{207}$$

Thus, by putting it all together, using a union bound to combine the probabilistic inequalities, we find that there exists constants $C_1, C_2, C_3$ such that, with probability at least $1 - \delta$,

$$L_{\mathcal{D}}(0, \mathcal{A}, \hat{h}) - L_{\mathcal{D}}(0, \mathcal{A}^*, h^*) \leq C_1 \sqrt{\frac{d_{\mathrm{VC}} \log\left(\frac{\sqrt{m}}{d_{\mathrm{vc}}}\right) + \log\left(\frac{\sqrt{m}}{\delta}\right)}{m}}$$

$$+ C_2\nu^{-1} \sqrt{\frac{d_N \log\left(\binom{N}{2}\frac{n\hat{n}}{d_N}\right) + \hat{n}d_{\mathrm{VC}} \log\left(\frac{n}{d_{\mathrm{vc}}}\right) + \log\left(\frac{\sqrt{n\hat{n}}}{\delta}\right)}{n\hat{n}}} + \epsilon, \tag{208}$$

where we note that the penalty terms arising from the union bound and converting between test and population losses have been absorbed using the constants.

$\square$

Thus, under the assumption of task diversity, we obtained an excess risk bound for a fixed target task, as was done in [5]. However, if we are interested in bounding the excess risk for a new, randomly drawn task, rather than a fixed target, task diversity is not necessary. In the following corollary, we demonstrate this by deriving an excess risk bound with respect to the population loss for a new, random task. While we only present a bound based on Corollary 6, similar excess risk bounds can be derived for the average case and from the other high-probability bounds.

**Corollary 8.** *Consider the setting of Corollary 6. Assume that $\mathcal{A}$ is an empirical risk minimizer, that $\mathcal{A}^*$ is an oracle algorithm, and let*

$$\hat{h} \in \arg\min_{h'} \hat{L}(1{:}\hat{n}, \mathcal{A}, h'), \tag{209}$$

$$h^* \in \arg\min_{h'} L_{\mathcal{D}}(-i, \mathcal{A}^*, h'). \tag{210}$$

*Then, there exists a constant $C$ such that, with probability at least $1 - \delta$ under $(Z, \hat{S}, S)$,*

$$L_{\mathcal{D}}(-i, \mathcal{A}, \hat{h}) - L_{\mathcal{D}}(-i, \mathcal{A}^*, h^*) \leq C \sqrt{\frac{d_N \log\left(\binom{N}{2}\frac{n\hat{n}}{d_N}\right) + \hat{n} d_{\mathrm{VC}} \log\left(\frac{n}{d_{\mathrm{VC}}}\right) + \log\left(\frac{\sqrt{n\hat{n}}}{\delta}\right)}{n\hat{n}}}. \tag{211}$$

*Proof.* We begin with the risk decomposition

$$L_{\mathcal{D}}(-i, \mathcal{A}, \hat{h}) - L_{\mathcal{D}}(-i, \mathcal{A}^*, h^*) = L_{\mathcal{D}}(-i, \mathcal{A}, \hat{h}) - \hat{L}(1{:}\hat{n}, \mathcal{A}, \hat{h}) + \hat{L}(1{:}\hat{n}, \mathcal{A}, \hat{h}) \tag{212}$$
$$- \hat{L}(1{:}\hat{n}, \mathcal{A}^*, h^*) + \hat{L}(1{:}\hat{n}, \mathcal{A}^*, h^*) - L_{\mathcal{D}}(-i, \mathcal{A}^*, h^*)$$
$$\leq L_{\mathcal{D}}(-i, \mathcal{A}, \hat{h}) - \hat{L}(1{:}\hat{n}, \mathcal{A}, \hat{h}) + \hat{L}(1{:}\hat{n}, \mathcal{A}^*, h^*) - L_{\mathcal{D}}(-i, \mathcal{A}^*, h^*),$$

where we used that $\hat{L}(1{:}\hat{n}, \mathcal{A}, \hat{h}) \leq \hat{L}(1{:}\hat{n}, \mathcal{A}^*, h^*)$. Next, by Lemma 5,

$$L_{\mathcal{D}}(-i, \mathcal{A}, \hat{h}) - \hat{L}(1{:}\hat{n}, \mathcal{A}, \hat{h}) + \hat{L}(1{:}\hat{n}, \mathcal{A}^*, h^*) - L_{\mathcal{D}}(-i, \mathcal{A}^*, h^*)$$
$$\leq \widetilde{L}(-1{:}\hat{n}, \mathcal{A}, \hat{h}) - \hat{L}(1{:}\hat{n}, \mathcal{A}, \hat{h}) + \hat{L}(1{:}\hat{n}, \mathcal{A}^*, h^*) - \widetilde{L}(-1{:}\hat{n}, \mathcal{A}^*, h^*) + 2\sqrt{\frac{\log(2\delta)}{2n\hat{n}}}. \tag{213}$$

This expression consists of the differences between the unobserved test losses and observed training losses for two different learning algorithms. We can thus use Corollary 6 to conclude that there exists a constant $C$ such that, with probability at least $1 - \delta$,

$$\widetilde{L}(-1{:}\hat{n}, \mathcal{A}, \hat{h}) - \hat{L}(1{:}\hat{n}, \mathcal{A}, \hat{h}) + \hat{L}(1{:}\hat{n}, \mathcal{A}^*, h^*) - \widetilde{L}(-1{:}\hat{n}, \mathcal{A}^*, h^*) + 2\sqrt{\frac{\log(2\delta)}{2n\hat{n}}}$$
$$\leq C \sqrt{\frac{d_N \log\left(\binom{N}{2}\frac{n\hat{n}}{d_N}\right) + \hat{n} d_{\mathrm{VC}} \log\left(\frac{n}{d_{\mathrm{VC}}}\right) + \log\left(\frac{\sqrt{n\tilde{n}}}{\delta}\right)}{n\hat{n}}}. \tag{214}$$

Here, the penalty term from the conversion between population and test loss has been absorbed into the constant $C$. From this, the desired result follows. $\qquad\square$