# OpenReview forum: "Evaluated CMI Bounds for Meta Learning: Tightness and Expressiveness"
_NeurIPS.cc/2022/Conference — NeurIPS 2022 Accept_

### Official Review · Reviewer_wciW · 2022-07-07

**Rating:** 6
**Confidence:** 4
**Soundness:** 3 good
**Presentation:** 3 good
**Contribution:** 2 fair

**Summary:**

The paper provides information-theoretic bounds for meta-learning with the recent e-CMI framework. At first, two square-root average e-CMI bounds are obtained with the two-step and one-step proof techniques, which are tighter than the previous corresponding information-theoretic bounds. Then the bounds are improved to faster rate decay ($\frac{1}{n\hat{n}}$) w.r.t the number of samples and number of tasks with the binary KL divergence. Next, the one-step and two-step e-CMI bounds are further extended to the high-probability setting. Finally, assuming the finite Natarajan dimension for the representation function set and finite VC dimension for the prediction function set, the expressiveness of e-CMI bounds is demonstrated to recover the scaling behaviors of meta-learning bounds obtained with classical learning theory.

**Questions:**

### suggestions
I suggest the author present Section 2 in a more logical way and simplify the notations if possible. For example, firstly introduce the original e-CMI framework that adopts supersample. Then to adapt the original e-CMI framework to meta-learning, the paper proposes the meta-supersample, which contains both ghost tasks and ghost task-specific samples, then introduce the membership, etc...

### Minor issues

1. Line 119 $\mathcal{R}^{\hat{n}}$
2. $\mathcal{F}$ undefined in 155
3. Appendix equation 28 and 29, $\gamma^2$ should be $\frac{\gamma^2}{2}$

**Limitations:**

The similarity and difference to previous Gaussian complexity bounds are mentioned when discussing the expressiveness for recovering the scaling behavior of classical learning theory.

**Strengths And Weaknesses:**

### Strengths
1. The paper presents theoretical results that bridge the information-theoretic meta-learning and the classical meta-learning theories.

2. The author first proposes the meta-learning bounds with the e-CMI framework, which are tighter than previous mutual information bounds.

3. The whole work is solid and the proof is clear.

### Weaknesses
1. The proof techniques are adapted from previous works, the one- and two-step bounds are based on previous proof ideas and the binary KL divergence bound is common in PAC Bayes theory.

2. The notations are a little cumbersome. Section 2 and the description for Figure 1 are redundant and hard to follow.

3. Lack of empirical validations.

---

> ### Author Response · Authors · 2022-08-02
> **Response to Reviewer wciW**
>
> We thank the reviewer for their constructive comments.
>
> ___________________________________________________________
> > _Weaknesses:
> The proof techniques are adapted from previous works, the one- and two-step bounds are based on previous proof ideas and the binary KL divergence bound is common in PAC Bayes theory._
>
> We acknowledge that the proofs build on powerful techniques from the literature, and tried to point this out in the paper.
> One of our novel contributions is to extend these techniques to the e-CMI setting.
> This requires careful handling of the random variables involved, and leads to results like Eq. (11), which has a distinctly different form from typical bounds based on the binary KL divergence.
> The same applies to the derivations of the bounds for classes with finite Natarajan and VC dimension.
> While previous work has only considered the expressiveness of information-theoretic bounds for the average case, we extend this analysis to high-probability bounds too.
>
> ___________________________________________________________
> > _Lack of empirical validations._
>
> While empirical evaluations are certainly an interesting avenue for future work, they are beyond the scope of this paper, the focus of which is to derive tighter information-theoretic bounds and determine the connection between them and generalization bounds from classical learning theory.
>
> ___________________________________________________________
> > _The notations are a little cumbersome. Section 2 and the description for Figure 1 are redundant and hard to follow. [...]
> I suggest the author present Section 2 in a more logical way and simplify the notations if possible. For example, firstly introduce the original e-CMI framework that adopts supersample. Then to adapt the original e-CMI framework to meta-learning, the paper proposes the meta-supersample, which contains both ghost tasks and ghost task-specific samples, then introduce the membership, etc..._
>
> While the nature of this work almost inevitably leads to somewhat cumbersome notation, we will try to simplify it further.
> For instance, we intend to remove superfluous hats wherever they appear.
> We thank the reviewer for their suggestions on how to improve the structure of Section 2, and will implement this suggested presentation order in the revised version of the paper.
> We have also corrected the issues that you point out in "Minor issues".

---

> > ### Comment · Reviewer_wciW · 2022-08-07
> > **Response**
> >
> > Thanks for the authors' reply. It does not change my assessment on the paper. I will keep the current score.

---

> > > ### Author Response · Authors · 2022-08-08
> > > **Thank you**
> > >
> > > Thank you for responding to the rebuttal, your detailed review of our work, and your helpful suggestions and comments.

---

### Official Review · Reviewer_7LdG · 2022-07-10

**Rating:** 6
**Confidence:** 4
**Soundness:** 3 good
**Presentation:** 3 good
**Contribution:** 3 good

**Summary:**

This paper concerns the problem of understanding the generalization of the meta learning algorithms. In particular, the authors study a “two-stage” algorithm in which in the first step the meta learner learns a meta hypothesis. Then, in the second stage, based on a limited number of samples, the base learner finds the output for a particular task.

This paper extends in a very interesting way the idea of conditional mutual information (CM) from  Steinke-Zakynthinou to the meta learning setup. In particular, they consider meta-supersample instead of only supersample as in the CMI paper. Then, they prove generalization bounds based on conditional mutual information in the same spirit as in the CMI paper. Also, they provide a high probability bound in Section 4.



**Questions:**

- One major question is why is the generalization gap important in meta learning? Note that the results of Tripuraneni et al concern the “excess risk” of the two-stage algorithm.

- The results in the paper do not give us any notion of task relatedness. It is opposed to the results in the transfer learning and meta learning literature in which we are mainly interested in answering when it is possible to do transfer learning.

- Can your results tell us when meta learning is not possible? Or when is the performance of the two-stage algorithm not good?

- The setup in the paper by Tripuraneni et al  is quite different from the general setup in the paper.

- Corollary 4 does not represent the power of the bounds. The bounds are worst-case bounds. As stated in the intro, we are interested in the algorithmic- and distribution- dependent results.

- The construction of the supersample was previously proposed in [35]. However, the bounds in this paper have better dependence on the number of samples. I think the authors should clarify the main technical differences between their proof idea and the proof idea in [35].

- What is the application of the bound to more practical algorithms such as MAML?



**Limitations:**

The authors did not discuss the limitations of their work in the paper.

**Strengths And Weaknesses:**


I think the most interesting aspect of the paper is providing an algorithm-dependent and distribution-dependent generalization bound for the meta learning problem. Also, the bounds are not loose as shown in Section 4.

Regarding weakness, the presentation of the additional structure to define the supersample structure should be improved.
Also, one of my concerns is that prior work on meta learning studies the “excess-risk”. However, in this work the authors study the generalization error. There is no discussion in the paper on why studying the generalization error instead of excess risk is interesting.
The next major weakness is that this work does not introduce any “task-relatedness” notion.

Also, the authors should spend more time on discussing the difference between their work and very related work of [35].

---

> ### Author Response · Authors · 2022-08-02
> **Response to Reviewer 7LdG, part 1**
>
> We thank the reviewer for their constructive comments.
>
> ___________________________________________________________
> > _Regarding weakness, the presentation of the additional structure to define the supersample structure should be improved._
>
> We are working on clarifying the presentation to make it as understandable as possible in the revised version.
> Following the suggestion of Reviewer wciW, we intend to improve it by first introducing the e-CMI framework for standard learning, and then the extension to meta learning.
>
> ___________________________________________________________
> > _Also, one of my concerns is that prior work on meta learning studies the “excess-risk”. However, in this work the authors study the generalization error. There is no discussion in the paper on why studying the generalization error instead of excess risk is interesting. [...]
> One major question is why is the generalization gap important in meta learning? Note that the results of Tripuraneni et al. concern the “excess risk” of the two-stage algorithm. [...]
> The next major weakness is that this work does not introduce any “task-relatedness” notion. [...] The results in the paper do not give us any notion of task relatedness. It is opposed to the results in the transfer learning and meta learning literature in which we are mainly interested in answering when it is possible to do transfer learning. [...] The setup in the paper by Tripuraneni et al. is quite different from the general setup in the paper._
>
> As the reviewer points out, there are two main discrepancies between the results in this paper and those of (Tripuraneni et al., 2020): their assumption of task diversity, which is absent from our analysis, and the study of excess risk for a specified task in their results, while we consider the generalization gap with respect to the test loss on a randomly drawn new task, as is done in, e.g., [25,27,29,30,31,33,35].
> These two points are highly related.
> Firstly, when worst-case, algorithm-independent bounds are used and applied to an empirical risk minimizer under a realizability assumption---as in (Tripuraneni et al., 2020)---excess risk bounds follow from generalization bounds, with an additional factor 2.
> This is achieved by using a risk decomposition (see the first equation on p. 13 in (Tripuraneni et al., 2020)), and applying the generalization bound once for the empirical risk minimizer and once for the optimal hypothesis.
> Since the generalization bound is worst-case and algorithm-independent, it applies identically for both hypotheses.
> The purpose of the task diversity assumption is to guarantee that the excess risk bound based on the training tasks always implies a bound for the excess risk on a fixed target task.
>
> What is done in (Tripuraneni et al., 2020) is roughly the following: Theorem 2 in (Tripuraneni et al., 2020) provides an excess risk bound for the specified task by performing a risk decomposition and applying a generalization bound in terms of Gaussian complexity within the specified task.
> However, the cost of using a suboptimal representation is still not explicitly bounded, and captured by the term $d_{\mathcal F,\mathcal F_0}( \mathbf{\hat h}, \mathbf{h^*})$.
> Next, the assumption of task diversity implies that, for some $(\nu,\epsilon)$,
>
> $d_{\mathcal F,\mathcal F_0}( \mathbf{\hat h}, \mathbf{h^*})\leq \bar d_{\mathcal F,f}( \mathbf{\hat h}, \mathbf{h^*})/\nu + \epsilon.$
>
> Here, $\bar d_{\mathcal F,f}( \mathbf{\hat h}, \mathbf{h^*})$ is the excess risk of the representation $\mathbf{\hat  h}$ on the tasks that are observed during training.
> This quantity, after a risk decomposition, is bounded by applying a Gaussian complexity generalization bound on the combined task and environment level (in our terminology, this is a one-step procedure).
>
> As it turns out, our generalization bounds can be readily converted into excess risk bounds with respect to the average population loss for a new task.
> Moreover, by using the same assumption of task diversity, we can also obtain the same kind of excess risk bounds for a specified task as is done in (Tripuraneni et al., 2020).
> We can achieve this by replacing the in-task Gaussian complexity generalization bound of (Tripuraneni et al., 2020) with an information-theoretic generalization bound for conventional learning, and the combined task and environment-level Gaussian complexity generalization bound with our new information-theoretic generalization bounds for meta-learning.
> For the case of hypothesis classes with bounded Natarajan or VC dimension, we can apply the techniques we present in the paper, for both the empirical risk minimizer and the oracle learning algorithm, to upper-bound the resulting information metrics.

---

> > ### Author Response · Authors · 2022-08-02
> > **Response to Reviewer 7LdG, part 2**
> >
> > Thus, to summarize, the reason that we did not initially consider excess risk bounds is that, for the risk decomposition step to work, it is crucial that the problem is realizable and that the algorithm is an empirical risk minimizer.
> > In contrast, all of the bounds that we present in the paper hold without such assumptions.
> > Furthermore, the reason that we did not use a task diversity assumption is that this is necessary only to bound the excess risk for a fixed task, but it is not necessary to bound the meta-test loss, i.e., the test loss on a randomly drawn new task.
> > In the revised version of the paper we will clarify these important points.
> >
> > Regarding the relevance of studying the generalization error rather than excess risk bounds, generalization bounds have the benefit of providing an explicit indication about the actual population loss that the learning algorithm will incur when used.
> > Excess risk bounds, however, characterize the difference between the population loss and the Bayes risk, i.e., the smallest attainable population loss, which in general is unknown.
> > Thus, excess risk bounds do not necessarily lead to direct statements about the population loss that a learning algorithm will lead to.
> >
> > As pointed out by the reviewer, many prior works on meta-learning and transfer learning consider the excess risk on a fixed target task, although as indicated above, there are many papers that bound the population loss.
> > We will therefore include a section at the end of the revised version of the paper presenting how such excess risk bounds can be obtained by using the information-theoretic bounds that we present in this paper.
> > Since the page limit for the main paper is still 9 pages during the rebuttal phase, this section is now in Appendix B in the supplementary material.
> > However, we intend to move it to the main paper in a revised version, when the page limit is increased to 10.
> > The derivation essentially follows the steps outlined above, with some additional technicalities that are necessary to deal with oracle algorithms.
> > We have uploaded a revision of the supplementary material where a full proof of this is given.
> > We thank the reviewer for highlighting this issue, as we believe that the resulting changes have greatly enriched the paper and increased its relevance with respect to prior work.
> >
> >
> > ___________________________________________________________
> > > _Can your results tell us when meta learning is not possible? Or when is the performance of the two-stage algorithm not good?_
> >
> > In this paper, we only study upper bounds on the generalization error.
> > While learning algorithms that lead to large information terms do not enjoy beneficial performance guarantees on the basis of our results, this does not strictly imply that they will not work.
> > To the best of our knowledge, tight information-theoretic lower bounds for learning are not available in the literature, even for conventional learning.
> >
> > ___________________________________________________________
> > > _Corollary 4 does not represent the power of the bounds. The bounds are worst-case bounds. As stated in the intro, we are interested in the algorithmic- and distribution- dependent results._
> >
> > As the reviewer correctly observes, the bounds that are given in terms of information measures are algorithm-dependent, while Corollary 4 presents algorithm-independent worst-case guarantees.
> > The purpose of this weakening is to demonstrate that the algorithm-dependent, information-theoretic bounds are powerful enough to capture these worst-case bounds.
> > However, the motivation for going beyond the worst-case bounds to pursue new, information-theoretic bounds is, as you point out, to be able to capture relevant properties of the learning algorithms and data distribution.

---

> > > ### Author Response · Authors · 2022-08-02
> > > **Response to Reviewer 7LdG, part 3**
> > >
> > > ___________________________________________________________
> > > > _Also, the authors should spend more time on discussing the difference between their work and very related work of [35]. [...]
> > > The construction of the supersample was previously proposed in [35]. However, the bounds in this paper have better dependence on the number of samples. I think the authors should clarify the main technical differences between their proof idea and the proof idea in [35]._
> > >
> > > The main differences between our work and the results in [35] are that
> > >
> > > 1. we consider e-CMI rather than ordinary CMI, leading to tighter bounds by the data-processing inequality,
> > > 2. while [35] only considers average bounds on the generalization gap, we also consider high-probability bounds and bounds that are given in terms of the binary KL divergence,
> > > 3. in addition to the two-step proof procedure of [35], we also use a one-step proof technique, leading to a better dependence on the number of samples,
> > > 4. we investigate the connection between the information terms that appear in our bounds and classical complexity measures, and
> > > 5. through a careful handling of the membership vector random variables, we disentangle them in the information terms that appear in our bounds to a greater extent than what is done in [35]. Specifically, the in-task membership vector appears as an argument of the environment-level information term in [35, Thm. 1], while it is conditioned on in Corollary 1 of our paper.
> > >
> > > We will extend the discussion in the paper to reflect the points above.
> > >
> > > ___________________________________________________________
> > > > _What is the application of the bound to more practical algorithms such as MAML?_
> > >
> > > While performing an in-depth study of MAML is beyond the scope of this paper, our results give some hints as to why and when MAML should work.
> > > First, since the MAML training objective requires that the meta-learned intialization parameter accommodates all training tasks reasonably well, one would not expect the inclusion/exclusion of individual tasks to have an outsized influence on the resulting losses.
> > > Similarly, since only a small number of gradient steps are to be taken within each task, the algorithm should also be stable with respect to the in-task samples---especially if the loss surfaces display flatness close to the selected initialization parameter.
> > > These considerations indicate that the e-CMI of the meta learner and base learner should be relatively small.
> > > So, while these arguments are somewhat speculative, they indicate that a more detailed study of MAML from the perspective of e-CMI generalization bounds may be fruitful.

---

> > > > ### Comment · Reviewer_7LdG · 2022-08-07
> > > > **Thank you**
> > > >
> > > > I would like to thank the authors for their detailed response. I will raise my score to 6.

---

> > > > > ### Author Response · Authors · 2022-08-08
> > > > > **Thank you**
> > > > >
> > > > > Thank you for your detailed evaluation of our work, your helpful comments and suggestions, and for raising the score.

---

### Official Review · Reviewer_9qK1 · 2022-07-12

**Rating:** 7
**Confidence:** 3
**Soundness:** 4 excellent
**Presentation:** 3 good
**Contribution:** 3 good

**Summary:**

This paper proves new information-theoretic generalization bounds for meta-learning, using evaluated conditional mutual information (e-CMI) which was introduced by Steinke and Zakynthinou (2020) who applied it in the context of standard supervised learning.

In Section 3. The paper gives distribution- and algorithm-dependent generalization bounds based on e-CMI. Then, they show that they can use these results to recover existing bounds in the literature using CMI. Additionally, they prove a generalization bounds based on binary KL divergence. This allows them to prove “fast rate” bounds in the optimistic setting, when interpolation is possible. They also prove high-probability bounds for these three results.

In Section 4. The authors instantiate these results and show that in a representation learning type of setup, their results recover existing upper bounds based on the complexity of the class used for representations and the class used for learning task-specific predictors, but the results in this paper also seem to be more general in the sense that they don’t require any assumptions.



**Questions:**

It might be good to clarify the learner and meta-learner that are used to establish Corollary 4, 5, and 6.

What’s the term: N choose 2, that appears in Corollary 4,5, and 6? It’s not clear to me what this is just from reading the corollary statements.


**Limitations:**

It might be good to reflect on the results of the paper and discuss their implications with respect to the current practice of meta-learning.


**Strengths And Weaknesses:**

The paper addresses an important problem of obtaining generalization guarantees for meta-learning. I think the results provided in this paper are nice and interesting, and in particular I liked how the generic results in Section 3 can be used to recover existing results in Section 4.

The paper is well written and easy to follow.

---

> ### Author Response · Authors · 2022-08-02
> **Response to Reviewer 9qK1**
>
> We thank the reviewer for their constructive comments.
>
> ___________________________________________________________
> > _Questions:
> It might be good to clarify the learner and meta-learner that are used to establish Corollary 4, 5, and 6._
>
> While the general bounds that we present involve information-theoretic quantities that depend on the specific learners that are used, Corollaries 4, 5 and 6 actually apply to any learner that outputs hypotheses from the specified classes.
> Indeed, the core of the proof relies only on information-theoretic properties of random variables defined on the space of possible losses for hypotheses from the given class.
> We will clarify this point in the revised version of the paper.
>
> ___________________________________________________________
> > _What’s the term: N choose 2, that appears in Corollary 4,5, and 6? It’s not clear to me what this is just from reading the corollary statements._
>
> Here, $N$ denotes the number of possible classes for the multiclass classification task, which, as you point out, was not stated explicitly in the submitted version of the paper.
> We will rectify this in the revised version.
>
> ___________________________________________________________
> > _Limitations:
> It might be good to reflect on the results of the paper and discuss their implications with respect to the current practice of meta-learning._
>
> The bounds that we derive hold for generic learning algorithms, and the upper bounds depend on properties of the algorithms that are used.
> This can be developed further to, for instance, use the bounds as regularizer terms in order to improve practical algorithms.
> As we demonstrate, the obtained bounds are also expressive enough to allow us to rederive some minimax generalization results.
> As discussed in the response to Review 7LdG, there are also indications that studying MAML from the perspective of e-CMI bounds may be fruitful.
> However, we do not provide any concrete recipe for these algorithmic improvements, although the generality, algorithm-dependence, and expressivity indicate that there is potential to develop one.
> We will include a discussion of these benefits and limitations of our analysis in the revised version of the paper.

---

> > ### Comment · Reviewer_9qK1 · 2022-08-08
> > **Thank you**
> >
> > Thank you for addressing and answering the questions I raised. I find the response satisfactory.

---

> > > ### Author Response · Authors · 2022-08-08
> > > **Thank you**
> > >
> > > Thank you for responding to the rebuttal, your detailed review of our work, and your helpful suggestions and comments.

---

### Meta-Review · Area_Chair_cYRX · 2022-08-25

**Recommendation:** Accept
**Confidence:** Certain

**Metareview:**

The reviewers agree that this is a solid contribution. Please do revise the paper according to the reviewers comments and the discussion.

**Award:**

No

---

### Decision · Program_Chairs · 2022-09-14

Accept